# UrbanMLLM: Joint Learning of Cross-view Imagery for Urban Understanding

## Abstract

Multimodal large language models (MLLMs) have exhibited remarkable capabilities for performing complex vision-language tasks in various domains. Currently, MLLMs based on urban imagery in urban studies are only developed focusing on remote sensing imagery. However, except for the macroscopic information from remote sensing imagery, effective urban understanding also requires detailed appearance information of urban zones from street-view imagery, which is largely overlooked by existing MLLMs. The primary challenges of developing such a versatile urban MLLM are twofold. Firstly, it needs a large-scale corpus with well-organized, cross-view urban imagery paired with corresponding text for cross-modal training. Secondly, traditional MLLMs typically learn image-text pairs independently, hard to support joint modeling of cross-view urban imagery. To address these challenges, in this work, we propose UrbanMLLM, a novel MLLM that jointly learns from remote sensing and street-view imagery to harness their complementary information. We first collect a large-scale dataset containing satellite-view and street-view imagery along with their geotags and annotated texts. Technically, we propose a brand MLLM architecture with a cross-view perceiver to explicitly connect visual information of cross-view urban imagery. We also introduce a novel pre-training paradigm based on structural interleaved urban image-text documents integrating satellite-view, street-view imagery and related textual descriptions. This approach encourages the model to implicitly learn the relationships between different types of urban imagery, enhancing the understanding in each domain. We evaluate our model on a comprehensive benchmark comprising 13 diverse urban understanding tasks across satellite-view, street-view, and cross-view domains. These tasks include scene classification, object reasoning, spatial relationship reasoning, geo-localization, landmark reasoning, and indicator prediction, providing a robust assessment of the model's capabilities. Extensive experiments demonstrate that UrbanMLLM achieves an average of 27.3% and 25.5% performance improvement compared with the best open-sourced and closed-sourced MLLMs, respectively. Moreover, we thoroughly study the impact of different pre-training data choices and model scales on performance, offering practical insights for effective MLLM design. The proposed UrbanMLLM offers a scalable and versatile solution for understanding urban environments.

## 1 Introduction

Urban imagery has been widely used for understanding cities in terms of urban spatial structure, functionality, and socio-economic status. The advancement in computer vision and multimodal learning has driven the utilization of multimodal urban data for urban understanding tasks, such as scene classification (Kuckreja et al., 2024; Mall et al., 2024), scene geo-localization (Vivanco Cepeda et al., 2024; Xu et al., 2024), urban indicator prediction (Fan et al., 2023; Hao et al., 2024), etc. More recently, benefiting from the impressive performance and generalizability of large language models (LLMs), multimodal large language models (MLLMs) have shown great potential for effectively solving distinct multimodal tasks in a "one-for-all" manner.

In the urban study area, there has been a line of works using MLLMs to tackle urban understanding tasks, while predominantly focusing on remote sensing (Kuckreja et al., 2024; Luo et al., 2024; Bazi

et al., 2024; Muhtar et al., 2024). Remote sensing imagery provides a macroscopic and comprehensive overview of how the city's functional zones are laid out while lacking some detailed contexts of the urban elements. Therefore, the existing MLLMs in urban studies can only deal with high-level urban understanding tasks such as land use classification and region captions. By comparison, street-view imagery captures a more fine-grained appearance of urban zones, providing complementary information such as building facades and heights. However, no existing MLLMs in urban science has explored integrating street-view imagery to enhance urban understanding. To enhance MLLMs' comprehensive understanding of urban environments, satellite imagery, which captures large-scale spatial layouts, and street-view imagery, which provides ground-level details, should be integrated and jointly learned within a unified model.

Achieving this goal needs to address two primary challenges. The first is the lack of well-organized multimodal urban data. Currently, publicly available urban datasets do not pair cross-view imagery with corresponding textual annotations, making them unable to support multimodal learning of MLLMs. The second challenge comes from the joint learning paradigm of connecting remote sensing and street-view imagery. In conventional MLLM frameworks (Liu et al., 2024c; Chen et al., 2024), visual features from cross-view imagery are encoded separately and aligned only with their respective annotated texts. These approaches fail to capture the complementary relationships between satellite and street-view imagery, leaving their information isolated.

In this work, we address the above two challenges and introduce a novel urban MLLM jointly learning from remote sensing and street-view imagery and associated textual data. Specifically, our efforts focus on both data contribution and methodology innovation. Given that existing publicly available datasets about urban imagery (Luo et al., 2024; Astruc et al., 2024) commonly lack paired cross-view images and large-scale annotated text information, we first collect a large-scale multi-modal urban imagery dataset. Our dataset covers the whole United States, comprising paired-up satellite-view and street-view images, together with geotags and annotated textual descriptions. We propose two key designs to break the visual knowledge isolation to facilitate the mutual learning of cross-view urban imagery. The first one is about the model architecture, where we propose a cross-view perceiver module that bridges the paired-up satellite-view and street-view visual features through a cross-attention mechanism. This design *explicitly* facilitates the exchange of information between the region-level context of satellite imagery and the fine-grained appearance details of street-view imagery. For example, the injection of street-view information to the satellite-view encoding can provide more detailed urban region context. The second part is a novel interleaved pre-training paradigm to enhance the mutual learning between cross-view imagery. In detail, we design coherent image-text documents that interleave the satellite-view image with matched street-view images and associated textual descriptions, forming a comprehensive profile of an urban region. Such interleaved training corpus helps MLLMs *implicitly* learn the relationship between different-view urban imagery via in-context learning. Through the explicit and implicit mutual learning between cross-view urban imagery, our proposed UrbanMLLM is expected to overcome the visual isolation issue, benefiting the comprehensive understanding of urban environments from diverse views.

For a comprehensive evaluation of urban understanding abilities for MLLMs, we build Urban-View Benchmark which includes 13 different tasks of urban perception (scene classification, geo-localization), reasoning (object reasoning, spatial relationship reasoning, landmark reasoning) and prediction (indicator prediction) based on single-view or cross-view urban imagery. Extensive experiments on the benchmark validate the noticeable superiority of UrbanMLLM on a wide range of urban understanding tasks, achieving an average 27.3% and 25.5% performance gain compared with the best open-sourced and closed-sourced MLLMs, respectively. Moreover, we study the impact of different pre-training data choices and model scales on the final performance, offering practical insights for effective MLLM design. This work serves as a foundational technique for addressing a wide range of urban-related tasks requiring comprehensive visual understanding capabilities. In brief, our major contributions can be summarized as follows:

- We propose a brand MLLM architecture with a designed cross-perceiver module to facilitate cross-fusion of the complementary visual context from satellite-view and street-view imagery.

- We construct a novel interleaved pre-training corpus that links satellite and street-view imagery through geo-location relationships, and propose a training paradigm that implicitly promotes mutual learning between cross-view imagery.

Table 1: Comparison of UrbanMLLM with other models in terms of data sources and targeted tasks.

| Method Type | Model | Data | | Task | | |
|---|---|---|---|---|---|---|
| | | Satellite Image | Street View Image | Perception | Reasoning | Prediction |
| CLIP-based | RemoteCLIP | ✓ | ✗ | ✗ | ✗ | ✓ |
| | UrbanCLIP | ✓ | ✗ | ✗ | ✗ | ✓ |
| | UrbanVLP | ✓ | ✓ | ✗ | ✗ | ✓ |
| MLLM-based | GeoChat | ✓ | ✗ | ✓ | ✓ | ✗ |
| | LHRS-Bot | ✓ | ✗ | ✓ | ✓ | ✗ |
| | SkysenseGPT | ✓ | ✗ | ✓ | ✓ | ✗ |
| | UrbanMLLM | ✓ | ✓ | ✓ | ✓ | ✓ |

- We establish a comprehensive benchmark including 13 different urban understanding tasks based on single-view or cross-view urban imagery. Extensive experiments verify that our model achieves substantial improvement in urban understanding over both open-source and closed-sourced MLLMs.

## 2 RELATED WORK

### 2.1 MULTIMODAL LARGE LANGUAGE MODELS (MLLMS)

With the rapid development and significant success of large language models (LLMs), recent research has focused on developing multimodal large language models (MLLMs) with the ability to comprehend both visual and textual knowledge, enabling them to address complex visual reasoning and understanding tasks. Existing MLLMs can be roughly divided into closed-source and open-source models. Closed-source MLLMs are mostly built based on corresponding commercial LLMs, such as GPT-4o (Achiam et al., 2023), Gemini (Reid et al., 2024) and Qwen-VL (Bai et al., 2023). These models benefit from the large-scale and extensive training corpus, which have been shown to exhibit powerful general multimodal understanding capabilities. By comparison, open-source MLLMs are usually smaller-scale, established by aligning a visual encoding branch to an off-the-shelf LLM. Following an earlier work LLaVA (Liu et al., 2024c), there are mainly two directions to advance the performance of MLLMs. The first line of works explores more advanced architectures such as introducing dual vision encoders (Li et al., 2024b), more sophisticated visual adapters (Cha et al., 2024) and the mixture-of-expert (MoE) strategy (Li et al., 2024c). Another line tries to uplift the performance of MLLMs with more beneficial pre-training data, such as interleaved multimodal data (Lin et al., 2024) and synthetic data (McKinzie et al., 2024). By comparison, our work introduces a novel MLLM architecture that integrates a cross-view perceiver module to enhance cross-view information fusion and contribute a unique interleaved pre-training corpus for MLLMs in urban areas.

### 2.2 MULTIMODAL MODELS FOR URBAN UNDERSTANDING

Understanding the urban environment usually requires multimodal information from diverse sources, such as satellite-view images, street-view images, POI information and geo-locations, etc. Existing methods in urban study can be categorized into two types: CLIP-based methods and MLLM-based methods, as shown in Table 1. From the data aspect, existing methods based urban imagery in urban study all focus on satellite images while overlooking using the street-view imagery for urban understanding. CLIP-based methods are mostly developed based on the contrastive learning strategy used in CLIP (Radford et al., 2021), such as training with satellite image-text pairs (Liu et al., 2024a; Yan et al., 2024), street-view image-text pairs (Hao et al., 2024) and satellite-view and street-view image pairs (Mall et al., 2024). These works can only deal with prediction tasks such as indicator prediction via end-to-end fine-tuning, but fail to conduct perception and reasoning tasks. Another line of research focuses on developing specialized MLLMs for problem-solving in the urban domain. Existing models, such as GeoChat (Kuckreja et al., 2024), SkysenseGPT (Luo et al., 2024), H$^2$RSVLM Pang et al. (2024), and EarthGPT (Zhang et al., 2024) only leverage remote sensing data including satellite images and annotated text for model learning. These models are capable of handling remote sensing perception and reasoning tasks but fail to deal with prediction tasks such as indicator predictions. However, relying solely on region-level knowledge is insufficient to capture

the complexities of urban environments, thereby limiting their applications for a wide range of urban understanding tasks. In contrast, our work proposes a novel learning paradigm based on cross-view urban image-text data which is capable of solving both remote sensing and street-view tasks.

# 3 METHODOLOGY

## 3.1 OVERVIEW

Traditional MLLMs are usually trained with paired-up separate image-text data, resulting in the knowledge isolation between different images. Such limitation constrains their ability of urban understanding, which requires a holistic comprehension of urban imagery from diverse perspectives. To overcome this issue, we propose two key designs to facilitate the comprehensive understanding on cross-view urban imagery. Firstly, we introduce a cross-view perceiver module in the MLLM architecture, explicitly enabling satellite-view and street-view visual contexts to complement each other. Secondly, we propose a novel interleaved pre-training paradigm leveraging structurally interleaved urban image-text contexts, integrating satellite-view and street-view imagery with corresponding textual descriptions. Training on such interleaved data enables MLLMs to learn relationships between cross-view urban imagery, leading to a more comprehensive understanding of urban environments. We elaborate the MLLM architecture enhanced with cross-view fusion in Section 3.2, followed by the designed pre-training paradigm based on interleaved urban contexts in Section 3.3.

## 3.2 CROSS-VIEW FUSION-ENHANCED URBANMLLM

Current MLLMs in urban studies primarily focus on remote sensing tasks. These models are typically developed by directly fine-tuning general-purpose MLLMs (e.g., LLaVA) on satellite image-text pairs. However, the effective urban understanding not only requires comprehending region-level knowledge from satellite-view imagery but also detailed contexts from street-view imagery. Unfortunately, this objective is unpromising to be achieved with the classical MLLM architecture, where images are individually encoded, failing to receive visual knowledge from relevant images.

Aiming to address the visual knowledge isolation issue, we introduce a *cross-view perceiver* module $g_\zeta(\cdot)$ to promote the awareness of urban imagery from other views during the visual encoding process. The cross-view perceiver is shown in the Figure 1. It performs 4 steps: (1) cross-attention from satellite-view image embedding (as queries) to the street-view image embedding; (2) cross-attention from street-view image embedding (as queries) to the satellite-view image embedding; (3) gating module before the residual connection; (4) MLP for aligning the semantic space of text. When both satellite-view and street-view images exist in the multimodal input, let $I_{st}$ denote a satellite-view image and $\{I_{sv}^i\}_{i=1}^n$ represent $n$ paired street-view images. The satellite-view and street-view images are firstly encoded by a pre-trained visual encoder $f_\phi(\cdot)$, resulting in visual features $\boldsymbol{f}_{st}$ and $\{\boldsymbol{f}_{sv}^i\}_{i=1}^n$, respectively. When encoding each street-view image $I_{sv}^i$, we inject the matched satellite-view features, giving rise to the fused feature $\boldsymbol{e}_{st \to sv}^i$ at the step of SI2SVI attn. For the satellite-view image, we first conduct an average-pooling on the visual features of $n$ paired street-view images then fuse it with the satellite-view feature, obtaining the fused feature $\boldsymbol{e}_{sv \to st}$ at the step of SVI2SI attn. Next, the fused visual feature is adaptively combined with the original feature using a gating strategy implemented with a one-layer MLP. The final visual embeddings $\mathbf{V}_{sv}^i$ and $\mathbf{V}_{st}$ are then obtained after a visual adapter (two-layer MLP). The whole operation of the cross-view perceiver is following:

$$\boldsymbol{e}_{st \to sv}^i = \text{MLP}(\text{Softmax}(\frac{\boldsymbol{f}_{sv}^i \boldsymbol{f}_{st}}{\sqrt{d_k}}) \boldsymbol{f}_{st}), \tag{1}$$

$$\mathbf{V}_{sv}^i = \text{MLP}(\text{Gating}(\boldsymbol{e}_{st \to sv}^i) + \boldsymbol{f}_{sv}^i), \tag{2}$$

$$\boldsymbol{e}_{sv \to st} = \text{MLP}(\text{Softmax}(\frac{\boldsymbol{f}_{st} \widetilde{\boldsymbol{f}}_{sv}}{\sqrt{d_k}}) \widetilde{\boldsymbol{f}}_{sv}), \tag{3}$$

$$\mathbf{V}_{st} = \text{MLP}(\text{Gating}(\boldsymbol{e}_{sv \to st}) + \boldsymbol{f}_{st}), \tag{4}$$

where $\widetilde{\boldsymbol{f}}_{sv} = \text{Pooling}(\{\boldsymbol{f}_{sv}^i\}_{i=1}^n)$.

If there's only single-view imagery in the multimodal input, the cross-view perceiver module receives two identical single-view images as input. In this way, the visual embedding fed to the LLM

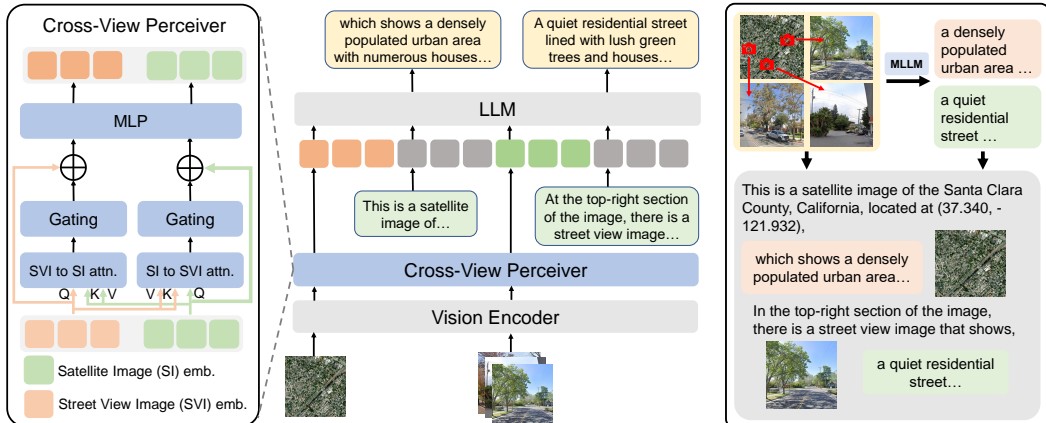

Figure 1: Architecture of the proposed UrbanMLLM. UrbanM-LLM employs cross-view perceiver to learn cross-view visual representations.

Figure 2: The pipeline of interleaved image-text data construction.

backbone is enhanced by the visual context from another view of the same urban region, which possesses more comprehensive urban context.

### 3.3 INTERLEAVED URBAN CONTEXT-BASED PRE-TRAINING

Existing explorations on general MLLMs have demonstrated that training on interleaved data yields superior performance than the traditional image-text pairs (Lin et al., 2024; McKinzie et al., 2024). The interleaved structure fosters semantic connections between multiple images, enabling MLLMs to better capture contextual relationships across images. This advantage aligns well with our objective of jointly learning from cross-view urban imagery to enhance the comprehensive understanding of urban visual knowledge. For instance, when predicting the geo-location of a satellite image, region-level visual information alone may be insufficient. Supplementing detailed street-view information can provide the necessary contextual knowledge for more accurate predictions.

Motivated by this, we introduce an urban context-based interleaved training paradigm tailored for urban understanding tasks. The core of this part is the construction of multimodal interleaved urban data as the training corpus. We first collect a large scale satellite-view and street-view imagery individually across the United States, and perform cross-view matching based on geotags (including located county, longitude and latitude), creating a paired cross-view urban imagery set $\mathcal{S} = \{(I_{st}, I_{sv}^1, I_{sv}^2, ..., I_{sv}^n) | n \in \mathbb{Z}^+\}$. We then employ an advanced MLLM InternVL (Chen et al., 2024) with carefully crafted prompts to efficiently generate textual descriptions for each image. Next, we link the cross-view images based on their geographical relationships and integrate their corresponding textual descriptions and geotags, forming a comprehensive urban profile for each element in $\mathcal{S}$. An illustrative example is shown in Figure 16. Training on such interleaved multimodal urban data benefits the MLLM to capture the relational knowledge between cross-view imagery, facilitating comprehensive urban understanding by fully integrating contextual information. Assuming that the interleaved document contains $K$ ordered urban images $\boldsymbol{I} = \{I_k\}_{k=1}^K$ interleaved with a $T$-length word sequence $\mathbf{w} = \{w_t\}_{t=1}^T$ tokenized by a $\theta$-parameterized LLM. The $k$-th image is successively processed by a frozen visual encoder $f_\phi(\cdot)$ and the cross-view perceiver $g_\zeta(\cdot)$ into $L$-length image tokens $\mathbf{V}_k = \{\mathbf{v}_l\}_{l=1}^L$. Denote $K(t)$ as the image index before the $t$-th word token. The pre-training objective of UrbanMLLM is to accurately predict the next word token with preceding image and word tokens:

$$\mathcal{L}(\Theta = \{\theta, \zeta\}, \mathbf{w}, \boldsymbol{I}) = -\mathbb{E}_t[\log p_\Theta(\mathbf{w}_t | \mathbf{w}_{<t}, \mathbf{V}_{<K(t)})], \quad \mathbf{V}_{K(t)} = g_\zeta \circ f_\phi(I_{K(t)}). \quad (5)$$

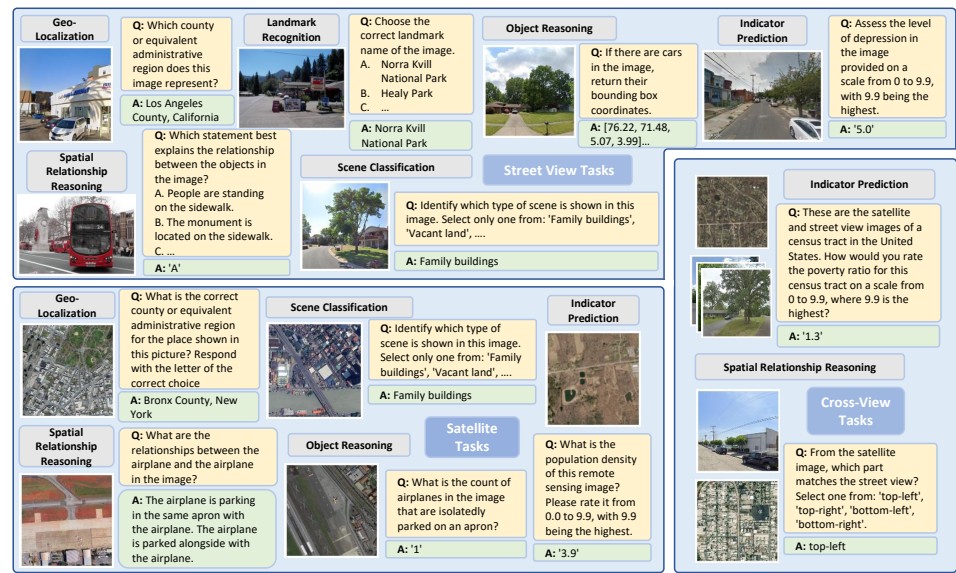

Figure 3: Examples of satellite, street view, and cross-view tasks in instruct tuning dataset. Diverse task categories include Scene Classification (SC), Object Reasoning (OR), Landmark Recognition (LR), Spatial Relationship Reasoning (SRR), Geo-Localization (GL) and Indicator Prediction (IP).

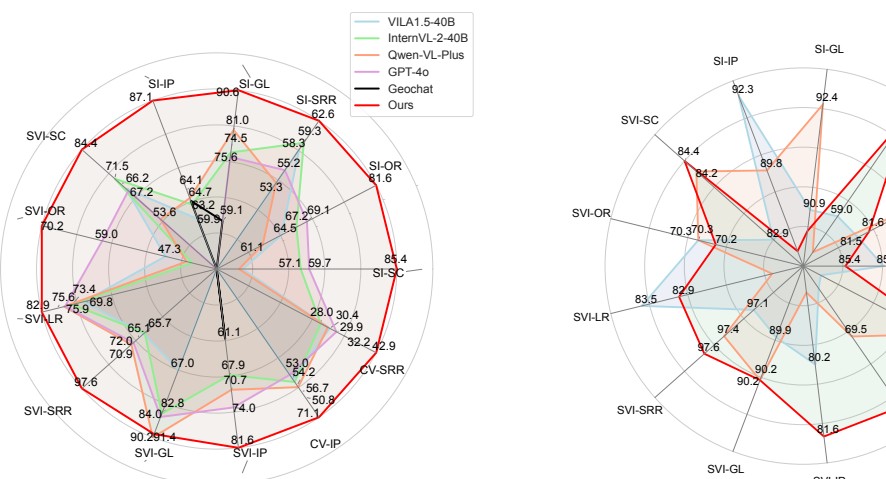

Figure 4: UrbanMLLM consistently improves the downstream task accuracy compared with both open-sourced and closed-sourced MLLMs. Abbreviations SI, SVI and CV stand for satellite imagery task, street view imagery task, cross-view task.

Figure 5: A performance comparison of UrbanMLLM's 3B, 8B, and 13B models across 13 urban understanding tasks.

## 4 EXPERIMENTS

In this section, we evaluate UrbanMLLM on three types of task: satellite view domain, street view domain and cross-view domain and then present the impact of various design choices on model performance.

### 4.1 EXPERIMENTAL SETUP

**Dataset** We use over 2 million satellite and street view images to build a large-scale cross-view interleaved pretraining dataset. Street view images offer ground-level details and appearance, while

Table 2: Satellite imagery-based urban understanding results on five tasks.

| Satellite Imagery Task | SC | | OR | SRR | GL | IP | |
|---|---|---|---|---|---|---|---|
| Sub-task | Single | Multi | | | | Pop | Nightlight |
| LLaVA-N-8B | 0.622 | 0.292 | 0.616 | 0.402 | 0.608 | 0.597 | Failed |
| LLaVA-OV-7B | 0.588 | 0.316 | 0.602 | 0.594 | 0.714 | 0.572 | Failed |
| CogVLM2-19B | 0.678 | 0.122 | 0.595 | 0.458 | 0.455 | 0.750 | Failed |
| LLaVA-N-34B | 0.574 | 0.220 | 0.629 | 0.588 | 0.608 | 0.597 | Failed |
| VILA1.5-40B | 0.650 | 0.152 | 0.645 | 0.583 | 0.475 | 0.599 | Failed |
| InternVL-2-40B | 0.664 | 0.479 | 0.672 | 0.593 | 0.756 | 0.632 | Failed |
| Qwen-VL-Plus | 0.589 | 0.191 | 0.611 | 0.533 | 0.810 | 0.647 | Failed |
| GPT-4o | 0.680 | 0.513 | 0.691 | 0.552 | 0.745 | 0.484 | Failed |
| GeoChat | 0.435 | 0.214 | 0.528 | 0.404 | 0.591 | 0.641 | Failed |
| LHRS-Bot | 0.439 | 0.128 | 0.568 | 0.386 | 0.243 | 0.533 | 0.449 |
| UrbanMLLM-3B | 0.901 | 0.816 | 0.815 | 0.590 | 0.909 | **0.923** | 0.735 |
| UrbanMLLM-8B | **0.910** | **0.825** | **0.821** | 0.577 | **0.924** | 0.898 | **0.789** |
| UrbanMLLM-13B | 0.898 | 0.810 | 0.816 | **0.626** | 0.906 | 0.871 | 0.728 |
| Improv. | 33.8% | 60.8% | 18.8% | 5.4% | 14.1% | 23.1% | - |

satellite imagery provide top-down views, capturing urban structures for comprehensive understanding of the entire landscape. Satellite and street view imagery in the same census tract are batched together with descriptive captions generated by MLLM. The corresponding county name and coordinates of the satellite image are also integrated into the batch.

We also construct an instruction tuning dataset for a variety of urban tasks, ranging from perception, reasoning to numerical prediction, as detailed below: **Satellite Imagery Tasks (SI):** Scene Classification (SC), Object Reasoning (OR), Spatial Relationship Reasoning (SRR), Geo-Localization (GL),Indicator Prediction (IP), population density prediction (Pop) and nightlight intensity prediction (Nightlight) are the sub-task of Indicator Prediction. Single Scene Classification (Single) and Multi-Scene Classification (Multi) are the sub-task of Scene Classification. **Street View Imagery Tasks (SVI):** Scene Classification (SC), Object Reasoning (OR), Landmark Recognition (LR), Spatial Relationship Reasoning (SRR), Geo-Localization (GL), Indicator Prediction (IP), predicting the beautiful (BF), wealthy (WE) and depressing (DP) level are the sub-tasks of Indicator Prediction. **Cross-View Tasks (CV):** Spatial Relationship Reasoning (SRR), Indicator Prediction (IP), predicting the median income (Med. income), poverty ratio (Pov. ratio), total population (Population) and depression rate (Depr. rate) level are the sub-tasks of Indicator Prediction. More details on the task settings and evaluation can be seen in A.6.

**Implementation** We initialize our model's weights using the pretrained VILA-1.5 model, and adapt the AdamW optimizer with a cosine learning rate scheduler during training. The training process consists of two stages: in the first stage, we train on the entire interleaved pretraining dataset with a batch size of 8 for one epoch, corresponding to 7200 steps with 8 hours. For the second stage, we fine-tune the model on the instruct tuning dataset at a batch size of 16 for one epoch with 8 hours. More information about baselines can be seen in A.5.

## 4.2 RESULTS

We compare the performance of our proposed UrbanMLLM with baselines on three tasks: satellite imagery task, street view imagery task and cross-view task on Table 2, Table 3 and Table 4. Based on these results, we have these noteworthy observations:

- **UrbanMLLM achieves the best performance across both satellite view and street view tasks.** The results showcase that UrbanMLLM achieves state-of-the-art performance, which successfully demonstrates the effectiveness of the proposed model for urban understanding tasks. We observe that our model achieves over 85% accuracy on simple perception tasks, such as scene classification and geo-localization, significantly outperforming general MLLMs. For more fine-grained tasks and object-level reasoning, our model outperformed the optimal baseline by 18.8%

Table 3: Street view imagery-based urban understanding results on six tasks.

| Street View Task | SC | OR | LR | SRR | GL | IP | | |
|---|---|---|---|---|---|---|---|---|
| Sub-task | | | | | | BF | WE | DP |
| LLaVA-N-8B | 0.513 | 0.492 | 0.643 | 0.705 | 0.575 | 0.600 | 0.593 | 0.413 |
| LLaVA-OV-7B | 0.694 | 0.572 | 0.711 | 0.742 | 0.782 | 0.678 | 0.778 | 0.627 |
| CogVLM2-19B | 0.500 | 0.460 | 0.580 | 0.222 | 0.642 | 0.482 | 0.435 | 0.385 |
| LLaVA-N-34B | **0.870** | 0.548 | 0.691 | 0.775 | 0.637 | 0.757 | 0.727 | 0.283 |
| VILA1.5-40B | 0.672 | 0.473 | 0.698 | 0.657 | 0.670 | 0.509 | 0.717 | 0.216 |
| InternVL-2-40B | 0.715 | 0.423 | 0.734 | 0.651 | 0.828 | 0.662 | 0.747 | 0.629 |
| Qwen-VL-Plus | 0.536 | 0.434 | 0.759 | 0.720 | **0.914** | 0.635 | 0.724 | 0.762 |
| GPT-4o | 0.662 | 0.590 | 0.756 | 0.709 | 0.840 | 0.824 | 0.723 | 0.673 |
| GeoChat | 0.316 | 0.378 | 0.282 | 0.279 | 0.306 | 0.577 | 0.605 | 0.652 |
| LHRS-Bot | 0.532 | 0.221 | 0.295 | 0.316 | 0.242 | 0.189 | 0.325 | 0.255 |
| UrbanMLLM-3B | 0.829 | **0.703** | **0.835** | 0.971 | 0.899 | 0.836 | 0.775 | **0.795** |
| UrbanMLLM-8B | 0.842 | **0.703** | 0.814 | 0.974 | 0.902 | 0.841 | 0.778 | 0.746 |
| UrbanMLLM-13B | 0.844 | 0.702 | 0.829 | **0.976** | 0.902 | **0.864** | **0.790** | 0.793 |
| Improv. | -3.0% | 19.0% | 10.0% | 25.9% | -1.3% | 4.9% | 1.5% | 4.3% |

Table 4: Cross view imagery-based urban understanding results on two tasks.

| Cross-View Task | IP | | | | SRR |
|---|---|---|---|---|---|
| Sub-task | Depr. rate | Med. income | Pov. ratio | Population | |
| LLaVA-OV-7B | 0.487 | 0.557 | 0.521 | 0.462 | 0.235 |
| VILA1.5-40B | 0.436 | 0.672 | 0.540 | 0.474 | 0.304 |
| InternVL-2-40B | 0.538 | 0.597 | 0.572 | 0.462 | 0.280 |
| Qwen-VL-Plus | 0.512 | 0.648 | 0.618 | 0.489 | 0.299 |
| GPT-4o | Failed | 0.684 | **0.848** | 0.499 | 0.322 |
| UrbanMLLM-3B | **0.759** | 0.790 | 0.804 | 0.588 | 0.389 |
| UrbanMLLM-8B | 0.653 | 0.773 | 0.760 | **0.596** | 0.421 |
| UrbanMLLM-13B | 0.714 | **0.798** | 0.762 | 0.571 | **0.429** |
| Improv. | 41.1% | 16.7% | -5.2% | 19.4% | 33.2% |

and 19.2%, respectively. This is because our model is pretrained on a large dataset of street-view and satellite images, allowing it to retain highly effective foundational image perception abilities. For more challenging reasoning tasks, such as predicting population density, our model outperforms the best general models by 23.1%, and surpassed the leading specialized models by 44.0% . It is important to note that in the SRR task, due to the limited availability of spatial relationship reasoning datasets in remote sensing, we used a task setup based on SkySenseGPT. As a result, our model has not previously encountered this specific task in the context of satellite imagery. Despite this, our model achieves performance comparable to general models, demonstrating its ability to acquire spatial relationship reasoning skills alongside its target inference capabilities.

On street-view tasks, our model achievs a 25.7% improvement in spatial relationship reasoning, a 6.5% increase in average prediction accuracy, and a 10.0% advantage in landmark recognition. Although its performance in geographic location prediction is slightly lower, trailing the closed-source model by 0.012, the model's consistency and strong results across other tasks demonstrate its overall robustness and effectiveness in street-view tasks. Through pre-training, the model develops a nuanced understanding of spatial relationships and world knowledge, as well as the ability to interpret abstract concepts such as beauty, wealth, or feelings of depression. Our model's performance on average metrics is comparable to that of the best open-source MLLMs, demonstrating its strong generalization ability in handling complex urban understanding tasks.

- **UrbanMLLM demonstrates greater consistency in cross-view tasks.** Table 4 showcases the performance of various models on cross-view tasks. It is evident that UrbanMLLM outperforms the baseline models in most tasks, especially in key areas such as depression rate, poverty ratio, and SRR, demonstrating their effectiveness in handling complex cross-view tasks. For exam-

ple, UrbanMLLM-3B achieves the best performance in the depression rate task, outperforming InternVL-2-40B by 41%, which is a significant improvement over other models. UrbanMLLM-8B excels in both median income and SRR, with the former showing a 15% improvement over the second-best model, VILA-1.5-40B, and the latter surpassing GPT-4o by 33.2%, highlighting its strong spatial reasoning capability. This indicates that larger models, such as UrbanMLLM-8B, are better suited for tasks that require complex spatial and economic reasoning. In contrast, other MLLMs like VILA-1.5-40B and Qwen-VL-Plus show mixed performance. While VILA performs relatively well on the median income task, it falls behind in other tasks. GPT-4o, despite excelling in the poverty ratio task, fails to complete the depression rate task, revealing a lack of consistency. In summary, UrbanMLLM provides more balanced and superior performance across multiple tasks, significantly outperforming the baseline models, which often exhibit strengths in specific areas but lack overall consistency.

- **Model size enhances performance but complexity of urban understanding task determines optimal gains.** Firstly, there is a clear trend that larger MLLMs, such as UrbanMLLM-13B, generally outperform smaller models like UrbanMLLM-3B across various tasks. This is demonstrated in Figure 5. For example, in the CV-SRR task, the 13B model achieves a score of 0.429, compared to 0.389 for the 3B model, indicating that increased model size often leads to better performance. Similar patterns are seen in tasks like object reasoning and spatial relationship reasoning, suggesting that larger MLLMs capture the complexities of image-based tasks more effectively, whether in single- or multi-task settings. However, in cross-view tasks (Table 4), the performance of the 3B and 8B models is nearly identical, with the 3B or 8B model even slightly outperforming the 13B model in the depression rate task. This indicates that MLLMs do not always guarantee superior performance, and that task complexity and data characteristics also significantly influence results.

## 4.3 EVALUATION ON URBANVIEW BENCHMARK

We evaluate our model with different open-source and closed-source MLLMs on our benchmark. We tested various models with the same set of questions on the same dataset. Due to differences in the models' ability to follow instructions, many do not provide answers exactly matching the ground truth but instead include additional explanatory text. Therefore, we consider a response correct as long as it contains the correct answer.

As shown in Table 2, 3 4 and Figure 4, our benchmark reveals the key challenges and limitations of current MLLMs in real-world urban environments. The results show that most advanced MLLMs do not perform well in satellite and street view tasks. For satellite view tasks, the top-performing closed-source models, such as GPT-4o and another leading model,InternVL-2-40B, achieved only 52.4% and 54.2% on average across various metrics. On street view imagery, their performance is similarly limited, with average scores of 72.2% and 67.4%, respectively. This discrepancy is because most of our images is collected recently, while the training data for these MLLMs generally lacks similar real-world data (Wang et al., 2024). Furthermore, many current MLLMs do not yet support multi-image inputs, and those do rarely handle tasks involving joint cross-view predictions for urban understanding. Consequently, this benchmark clearly highlights the limitations of advanced models, showing their challenges in performing well on urban understanding tasks, especially with street view and remote sensing images, and in joint cross-view prediction tasks.

## 4.4 ABLATION ANALYSIS

To evaluate the effectiveness of each module in UrbanMLLM, we evaluate the performance of various task of different model variants in Table 5, Table 6, and Table 7. Specifically, we evaluate the UrbanMLLM without cross-view perceiver (w/o Perceiver), satellite imagery in the pretraining stage and cross-view perceiver (w/o SI+Perceiver) , street view imagery in the pretraining stage and cross-view perceiver (w/o SVI+Perceiver), satellite imagery and street view imagery in the pretraining stage and cross-view perceive (w/o SI+SVI+Perceiver). Note that in the variant without the cross-view Perceiver (w/o Perceiver), a two-layer MLP is implemented as a replacement.

According to the results, cross-view perceiver is the most essential module for explicitly facilitate mutual learning of cross-view urban imagery. It brings 2%-81% gains for all tasks, because satellite and street view images represent two completely different modalities of information, making it difficult to directly integrate and interact within LLMs during the pretraining stage to learn cross-

Table 5: Ablation study of UrbanMLLM variants on satellite imagery tasks.

| Variants | SC | | OR | SRR | GL | IP | |
|---|---|---|---|---|---|---|---|
| | Single | Multi | | | | Pop | Nightlight |
| UrbanMLLM-8B | **0.910** | **0.825** | **0.821** | 0.577 | **0.924** | 0.898 | **0.789** |
| w/o Perceiver | 0.749 | 0.596 | 0.732 | 0.106 | 0.427 | 0.869 | 0.747 |
| w/o SI+Perciver | 0.897 | 0.819 | 0.814 | 0.549 | 0.921 | **0.913** | 0.737 |
| w/o SVI+Perceiver | 0.907 | 0.822 | 0.818 | **0.615** | 0.919 | 0.880 | 0.707 |
| w/o SI+SVI+Perceiver | 0.888 | 0.806 | 0.818 | 0.604 | 0.903 | 0.869 | 0.713 |

Table 6: Ablation study of UrbanMLLM variants on street view imagery tasks.

| Variants | SC | OR | LM | SRR | GL | IP | | |
|---|---|---|---|---|---|---|---|---|
| | | | | | | BF | WE | DP |
| UrbanMLLM-8B | 0.842 | **0.703** | **0.814** | **0.974** | **0.902** | 0.841 | 0.778 | 0.746 |
| w/o Perceiver | 0.666 | 0.640 | 0.455 | 0.860 | 0.642 | 0.771 | 0.760 | 0.727 |
| w/o SI+Perceiver | 0.829 | 0.699 | 0.805 | **0.974** | 0.891 | **0.897** | **0.786** | **0.762** |
| w/o SVI+Perceiver | **0.844** | 0.701 | 0.814 | 0.973 | 0.887 | 0.880 | 0.776 | 0.754 |
| w/o SI+SVI+Perceiver | 0.772 | 0.696 | 0.812 | 0.964 | 0.888 | 0.878 | 0.774 | 0.737 |

Table 7: Ablation study of UrbanMLLM variants on cross-view tasks.

| Variants | IP | | | | SRR |
|---|---|---|---|---|---|
| | Depr. rate | Med. income | Pov. ratio | Population | |
| UrbanMLLM-8B | 0.653 | 0.773 | **0.760** | **0.596** | **0.421** |
| w/o Perceiver | 0.495 | 0.462 | 0.520 | 0.478 | 0.247 |
| w/o SI+Perceiver | **0.752** | 0.792 | 0.759 | 0.520 | 0.372 |
| w/o SVI+Perceiver | 0.714 | 0.782 | 0.759 | 0.531 | 0.419 |
| w/o SI+SVI+Perceiver | 0.674 | **0.793** | 0.697 | 0.557 | 0.348 |

view semantic information. Therefore, a specialized mechanism is required to fuse these modalities in advance. The use of satellite image data during pretraining has a significant impact on satellite image-related tasks, contributing performance gains ranging from 0.3% to 7.1%. However, its effect on various economic indicators in cross-view tasks differs. For example, it resulted in a 15.2% improvement in the depression rate task but caused a 12.8% decrease in accuracy for total population estimation of one region. This difference is due to the depression rate being more closely related to visible green space in satellite images, while population estimation requires a more nuanced understanding of urban environmental factors. Similarly, using street view image data during pretraining has a greater impact on street view-related tasks compared to satellite data, contributing performance gains of 0.3% to 1.7%. This demonstrates that pretraining with data closely aligned to downstream tasks can significantly enhance model performance. Additionally, the interleaved image-text pretraining on satellite and street view images provides a task-agnostic yet semantically rich initialization, contributing a performance gain of 0.3% to 9% in tasks such as scene classification on street view images. Therefore, the interleaved image-text pretraining of the two types of images, along with the cross-view perceiver, are essential components of our approach.

# 5 CONCLUSION

In this paper, we propose UrbanMLLM, a novel multimodal large language model designed to jointly learn from remote sensing and street-view imagery for comprehensive urban understanding. By leveraging a large-scale cross-view dataset and a cross-view perceiver architecture, UrbanMLLM effectively captures complementary information from satellite-view and street-view. Our model outperforms existing MLLMs, achieving significant improvements across various urban understanding tasks.

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

# A APPENDIX

## A.1 ETHICS

Our model uses a large amount of satellite and street view images, which poses a potential risk to individual privacy. While the resolution of the satellite imagery is not high enough to identify individuals, it can still detect environmental changes resulting from human activity. The street view images were crawled and downloaded from the Google platform, with key information blurred, ensuring that no private information is compromised.

Our model is designed to better understand cities from cross views by incorporating various data types to enhance its understanding capabilities. To minimize the misuse of our model and data, we will release the dataset and trained model only to those who agree to adhere to ethical guidelines. By following these guidelines, users agree to comply with laws, regulations, and the ASPRS Code of Ethics. It is also important to note that the data used for this training is already freely available and public, so our model does not exacerbate privacy concerns.

## A.2 LIMITATIONS AND FUTURE WORK

Although our model, UrbanMLLM, covers a wide range of urban understanding tasks and achieves state-of-the-art performance, there are still some limitations. Our dataset is limited to the United States, and the generalization of the model to other countries may require additional data collection and pre-training. Therefore, we plan to extend the dataset to cover more regions and improve the general applicability of the model, and explore the use of additional modalities for urban understanding tasks.

## A.3 IMPLEMENTATION DETAILS FOR REPRODUCIBILITY

We perform experiments using Python 3.10 and Pytorch 2.3.0+cu121 with $8\times$ NVIDIA A100 GPUs. Here we provide detailed values of the hyper-parameters used in the experiments for reproducibility in Table 13 and Table 14 for the training and testing, respectively.

## A.4 EXPERIMENTAL RESULTS

### A.4.1 ABLATION STUDY

As our model supports multi-image as input, we conduct an ablation study on the number of images (N = 2, 4, 6) for indicator prediction tasks. For example, 6 images means one satellite image and five street-view images as input. The results are as follows ( 8). It can be seen that more images bring certain performance gain for the indicator prediction task.

Table 8: Image number ablation study of UrbanMLLM on cross-view imagery tasks.

| Number of input images | IP | | | |
|:---:|:---:|:---:|:---:|:---:|
| | Depr. rate | Med. income | Pov. ratio | Population |
| 2 | 0.724 | 0.766 | 0.696 | 0.596 |
| 4 | 0.772 | 0.789 | 0.689 | 0.596 |
| 6 | 0.764 | 0.809 | 0.720 | 0.614 |

### A.4.2 DATA SCALE STUDY

We also provide the results of UrbanMLLM trained on different dataset scales, including 0.35 million, 0.92 million, and 1.86 million images. The results (Table 9, Table 10, Table 11) show that the performance of UrbanMLLM improves a little with the increase in dataset scale. Because our pre-training data is much less than data size that scaling law requires.

Table 9: Satellite imagery-based urban understanding results with different data scale on five tasks.

| Satellite Imagery Task | SC | | OR | SRR | GL | IP | |
|---|---|---|---|---|---|---|---|
| Sub-task | Single | Multi | | | | Pop | Nightlight |
| 0.35M | 0.899 | 0.816 | 0.822 | 0.644 | 0.921 | 0.904 | 0.741 |
| 0.92M | 0.908 | 0.825 | 0.823 | 0.608 | 0.935 | 0.923 | 0.775 |
| 1.86M | 0.910 | 0.825 | 0.821 | 0.577 | 0.924 | 0.898 | 0.789 |

Table 10: Street view imagery-based urban understanding results with different data scale on six tasks.

| Street View Imagery Task | SC | OR | LR | SRR | GL | IP | | |
|---|---|---|---|---|---|---|---|---|
| Sub-task | | | | | | BF | WE | DP |
| 0.35M | 0.840 | 0.706 | 0.847 | 0.982 | 0.902 | 0.834 | 0.777 | 0.746 |
| 0.92M | 0.835 | 0.703 | 0.825 | 0.970 | 0.894 | 0.838 | 0.761 | 0.773 |
| 1.86M | 0.842 | 0.703 | 0.814 | 0.974 | 0.902 | 0.841 | 0.778 | 0.746 |

Table 11: Cross view imagery-based urban understanding results with different data scale on two tasks.

| Cross-View Imagery Task | IP | | | | SRR |
|---|---|---|---|---|---|
| Sub-task | Depr. rate | Med. income | Pov. ratio | Population | |
| 0.35M | 0.754 | 0.781 | 0.744 | 0.486 | 0.415 |
| 0.92M | 0.701 | 0.805 | 0.750 | 0.555 | 0.418 |
| 1.86M | 0.653 | 0.773 | 0.760 | 0.596 | 0.421 |

### A.4.3 EVALUATION ON CITYBENCH

We also provide the results of the proposed dataset on other benchmarks, including Citybench (Feng et al., 2024). We select CityInfer, LocInfer, and Population as tasks to evaluate the performance. We use Accuracy, Accuracy@25km, and $R^2$ as evaluation metrics. More details can be found in the Citybench. The results are shown in Table 12. The proposed dataset outperforms the state-of-the-art models on these benchmarks, demonstrating the effectiveness of the proposed dataset for urban understanding tasks.

Table 12: Best Performance on Close-Source Model, Open-Source Model, and UrbanMLLM on Citybench.

| Model | CityInfer | LocInfer | Population |
|---|---|---|---|
| SOTA closed-source model | 0.862 | 0.797 | 0.122 |
| SOTA open-source model | 0.574 | 0.555 | -0.113 |
| UrbanMLLM-8B | 0.904 | 0.840 | 0.324 |

### A.4.4 CASE STUDY

We have added a bad case analysis in the revised paper. We show some examples of bad cases in scene classification and indicator prediction tasks. The results are shown in Figure 6, 7, 8. Firstly, in the scene classification task, the model misclassifies the image with a truck parking as a car parking. Although there are a few differences between the two classes, the more granular understanding of the urban environment is required to distinguish them. Secondly, in the indicator prediction task, as shown in Figure 7, 8, the model predicts the population density of an urban area as 6.8 and the actual value is 9.9 using a satellite image. The model fails to capture detailed information with a single-view image, which makes it challenging for the model to learn from the limited dataset. For poverty rate prediction, the model gets a high score of 5.4, but the poverty rate is 2.6. It's may be

```
prompt ='''
Classify the given image into the following classes.
Classes: taxiway, bridge, boarding_bridge, car_parking,
truck, containment_vessel, apron, smoke,
engineering_vehicle, goods_yard, truck_parking, gas_station,
unfinished_building, roundabout, storehouse, substation,
arch_dam, flood_dam, chimney, intersection, tank, airplane,
gravity_dam, runway, genset, tennis_court, boat,
basketball_court, breakwater, stadium. \nAnswer with all
applicable classes separated by commas.
'''
answers = "truck_parking, car_parking"

prompt ='''
Classify the given image into the following classes.
Classes: storehouse, smoke, ship, foundation_pit, crane,
gravity_dam, containment_vessel, cement_concrete_pavement,
dock, tennis_court, car_parking, roundabout,
unfinished_building, stadium, boat, cooling_tower,
intersection, car, apron, truck, ship_lock, tower_crane,
goods_yard, taxiway, arch_dam, tank. \nAnswer with all
applicable classes separated by commas.
'''
answers = "car_parking, car"
```

**Ground Truth:**
**truck_parking**

**Ground Truth:**
**car_parking, car**

Figure 6: Bad case on scene classification.

```
prompt ='''
Please rate the population density of this image from 0.0
to 9.9, with 9.9 being the highest. Only output the score
'X.X'.
'''
answers = "6.8"
```

**Ground Truth:**
**9.9**

```
prompt ='''
Assess the level of wealth in the image provided on a scale
from 0 to 9.9, with 9.9 being the highest. Only output the
score. Example format: 'X.X'.
'''
answers = "3.3"
```

**Ground Truth:**
**8.0**

Figure 7: Bad case on indicator prediction task in single view imagery understanding.

due to the number of street view images in the dataset is not enough to learn the detailed information of the urban environment.

Table 13: Hyperparameter settings for training.

|  | Stage1 | Stage2 |
| --- | --- | --- |
| Optimizer | AdamW | AdamW |
| Learning Rate | 5e-5 | 1e-4 |
| Batch Size | 8 | 16 |
| Accumulation Step(s) | 1 | 2 |
| Weight Decay | 0.0 | |
| Epoch(s)/Step(s) | 1 Epoch | 1 Epoch |
| Save Steps | 1200 | 750 |
| Scheduler | Cosine | |
| Warmup Ratio | 500 | 100 |
| Model Max Length | 2048 | |

```
prompt ='''
You're looking at satellite and street view images from a
census tract in the U.S. How would you score the poverty
ratio on a scale of 0 to 9.9, with 9.9 being the highest?
Provide only the number. Example format: 'X.X'.
'''

answers = "5.4"
```
**Ground Truth:2.6**

Figure 8: Bad case on indicator prediction task in cross view imagery understanding.

Table 14: Hyperparameter settings for testing.

| Hyper-parameter | Value |
|---|---|
| Temperature | 0.2 |
| Top_p | None |
| Num Beams | 1 |
| Conv Mode | v1/llama_3 |
| Max New Tokens | 128 |

## A.5 BASELINES

We evaluated several advanced MLLMs on the UrbanView Benchmark. However, some of the MLLMs are not pretrained on multi-image data or does not support multi-image inference, such as LLaVA-Next (Liu et al., 2024b) and CogVLM2 (Wang et al., 2023), so only single-image tasks are evaluated. For VILA-1.5 (Lin et al., 2023), InternVL2 (Chen et al., 2023), and LLaVA-OneVision (Li et al., 2024a), the whole benchmark evaluation is done. In addition to the open-source models, state-of-the-art closed-source models Qwen-VL-Plus and GPT-4o are also fully evaluated on the benchmark. Specifically, we assess a satellite domain-specific model, GeoChat (Kuckreja et al., 2024), to further prove our capability. To ensure fairness, domain-specific models that are not yet open-source are excluded from this evaluation.

## A.6 URBANVIEW DATASET AND BENCHMARK

As the thriving development of satellite and street view data, a series of publicly available datasets about urban imagery has been brought up. However, considering the complexity and diveristy of urban environment, single-view data of satellite-view or street-view is not enough. Therefore, to enhance MLLM's comprehensive and all-level understanding of cities, we propose UrbanView, a dataset and benchmark that composes of massive amount of multi-source and cross-view urban

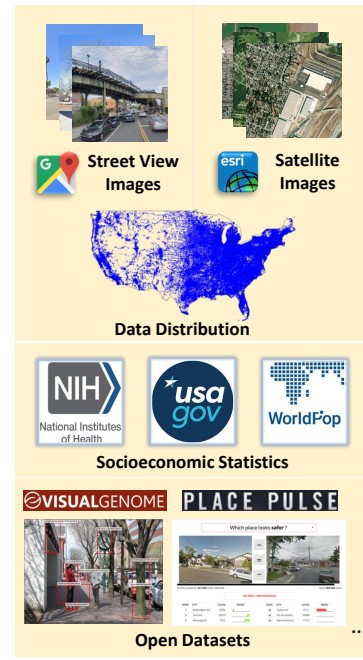

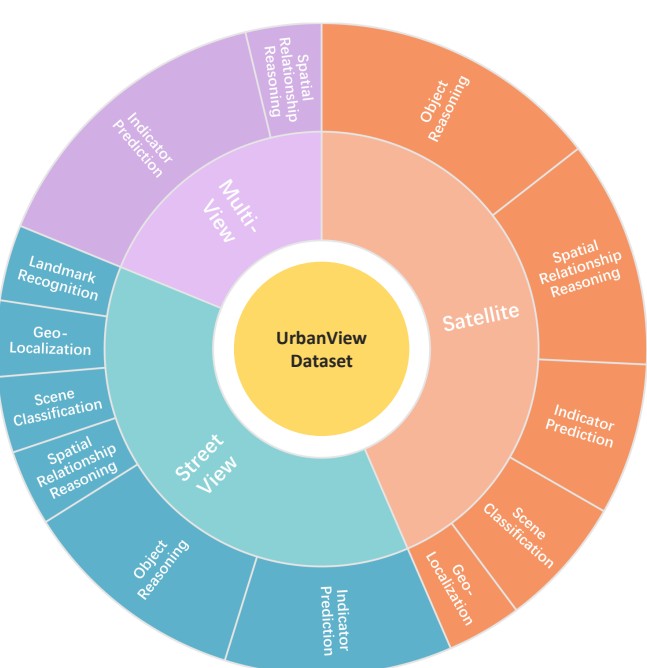

Figure 9: Data Collection          Figure 10: UrbanView Dataset Statistics.

imagery, in combination with various collected labels across geo-locations, grounded objects, spatial relationships, income and health indicators.

### A.6.1 DATA COLLECTION

The images are primarily collected from two sources: Google Maps API (Google, 2024) for street view images, and ESRI for satellite imagery. For street view imagery collection, we randomly generate 2,000 random points in each census tract polygon and use their coordinates to query Google Maps API, returning street view image patches and real coordinates. We scrape over 2 million street view images and all satellite imagery of zoom level 15 across the United States. We further gather a variety of socioeconomic data of census tract and grid level from world pop, NIH and US government. We also collect a series of open datasets, such as Google Landmarks Recognition Weyand et al. (2020), Visual Genome Krishna et al. (2016) and Place Pules Dubey et al. (2016) etc. By applying some domain-specific adaptation to the original ones, we build a more well-rounded UrbanView dataset.

The data used for dataset construction have a sparse yet overall coverage of the United States as shown in Figure 9. We use census tract boundary data in 2019 (Bureau, 2019), and gather street view and satellite images in 71,433 out of all 73,868 census tracts in the United States, which is about 96.7%. The Google street view images can be acquired using coordinate queries, however, we don not know the exact coordinate of where the street view exists. We randomly generate 2,000 points in each census tract and use these points to query street view images. This is a random process, thus we are not able to sample all the images in a census tract considering the time cost. In fact, in some less populated areas, it is quite hard to get street view images because the randomly generated query points in these areas are always off-road, which is also the main reason for the missing 3.3% coverage. In the end, we randomly sample about 200 images in each census tract, which have been proved to be effective in indicator prediction tasks.

The data size for each task of the UrbanView dataset and benchmark is listed below in Table 15:

Table 15: Dataset and benchmark data size for different sources and tasks

| Source | Task | Dataset Size | Benchmark Size |
|---|---|---|---|
| Street View | Scene Classification (SC) | 30,000 | 1,000 |
| | Object Reasoning (OR) | 90,000 | 1,000 |
| | Landmark Recognition (LR) | 30,000 | 1,000 |
| | Spatial Relationship Reasoning (SRR) | 30,000 | 1,000 |
| | Geo-Localization (GL) | 30,000 | 1,000 |
| | Indicator Prediction (IP) | 90,000 | 3,000 |
| Satellite | Scene Classification (SC) | 51,759 | 8,668 |
| | Object Reasoning (OR) | 115,115 | 5,556 |
| | Spatial Relationship Reasoning (SRR) | 90,000 | 8,250 |
| | Geo-Localization (GL) | 29,629 | 1,000 |
| | Indicator Prediction (IP) | 60,000 | 2,000 |
| Cross-View | Spatial Relationship Reasoning (SRR) | 30,000 | 1,000 |
| | Indicator Prediction (IP) | 120,000 | 4,000 |

### A.6.2 DATASET STRUCTURE AND CONSTRCUTION

We first build a large-scale cross-view interleaved pretraining dataset. For each census tract, we match the coordinate between satellite and street view imagery. Since we collect all satellite images in the United States, each street view image can find a match. However, in order to control the size of inputs, at most 5 street view images are matched with single satellite image. For the next step, we use a powerful open-source MLLM, InternVL2-40B, to generate detailed descriptive captions for them. Using a similar data structure in MMC4 (Zhu et al., 2024), the county name and coordinates of the satellite image are also embedded to the interleaved pre-training data, together with imagery and caption embeddings.

We use a Human-AI mixture method for pre-training caption quality validation. We first use two powerful open-source MLLM, VILA-1.5-40B and LLaVA-Next-34B to judge if the caption matches with the image. If either of them thinks it is not a match, we will proceed to send this case to GPT-4o, which has state-of-the-art comprehension ability, but not quite affordable for large-scale deployment. If GPT-4o also thinks there is a problem with the case, graduate-level human-being will manually check this case to give the final judgement. In order to quantify the caption quality improvement, we further use GPT-4o to regenerate captions for excluded images with human assistance to test the quality improvement. We use CLIP-Score as the evaluation metric and calculate our original caption score and cleaned caption score of 10,000 samples, resulting 29.99535 and 29.99571 respectively. As a matter of fact, the original caption quality is good enough and only about 1.3 out of a thousand images is marked as unmatched by two-stage MLLM verification, and the regeneration process enhances the caption quality only by 0.0012%.

Then we construct the instruct tuning dataset, which is categorized into 3 major types: satellite-only, street view-only, and cross-view data. Street view images offer ground-level data of various environments, including urban and rural areas. These images provide detailed features of the environment. In contrast, satellite imagery complements this by providing top-down views, capturing a overall perception of the entire landscape. In UrbanView dataset, not only satellite and street view images linked respectively with diverse tasks such as scene classification, object reasoning, spatial relationship reasoning, cross-view combinational tasks of socioeconomic prediction and image retrieval are delicately designed to further enhance MLLM's comprehensive understanding of urban environment.

In the construction of UrbanView dataset, a lot of street view tasks are in lack of groundtruth labels. Therefore, we use light-weight object detection specialized model to generate groundtruth bounding boxes for object reasoning tasks, and use powerful open-source MLLMs to identify the scene class and spatial relationship of the street view image. For geo-localization tasks, we simply use latitude and longitude of the images to match the boundaries of census tracts in the United States.

We construct an instruction tuning dataset for a variety of urban tasks, ranging from perception, reasoning to numerical prediction, as detailed below:

• **Satellite Tasks:**

The satellite Scene Classification (SC), Object Reasoning (OR), and Spatial Relationship Reasoning (SRR) dataset are the same as FIT-RS dataset in SkySenseGPT, since they have built a high-quality dataset and proved to be effective on satellite tasks. **Scene Classification (SC)**: Select which scene or scenes does this image conform to.

**Object Reasoning (OR)**: Respond the location, presence or count of a specific object.

**Spatial Relationship Reasoning (SRR)**: Select the correct object relationship displayed in the image.

**Geo-Localization (GL)**: Select which county this image belongs to. We use the latitude and longitude coordinate of each image to find the corresponding county it belongs to. Only the most populated 100 counties in the United States are taken account of. We also use multiple choices format for this task type, and the distraction choices are randomly chose from the 100 counties.

**Indicator Prediction (IP)**: Predict population density or nightlight intensity from 0.0 to 9.9. We follow the normalization method used in GeoLLM (Manvi et al., 2023), scaling down the population and nightlight density to the range of 0.0 to 9.9, and ask the MLLMs to give a direct estimation in this range. The population density data are sourced from WorldPop (Tatem, 2017) and the nightlight data are sourced from VIIRS (Li et al., 2020).

• **Street View Tasks:**

**Scene Classification (SC)**: Select which scene does this image conform to. The ground-truth is obtained using LLaVA-Next-34B, which have been verified to generate a pretty reasonable result on scene classification task.

**Object Reasoning (OR)**: Respond the location, presence or count of a specific object. There are three kinds of sub-tasks in object reasoning, all ground-truth annotations are generated by Grounding DINO (Liu et al., 2023), which has shown state-of-the-art ability on open vocabulary object detection. We further process the bounding box and object name results given by Grounding DINO to build object presence and counting dataset.

**Landmark Recognition (LR)**: Select the correct landmark name shown in the image. We use images from google landmarks dataset v2, and select the street view images in the dataset via LLaVA-Next-34B. Multiple choice questions are made based on the correct landmark name and three distraction landmark names.

**Spatial Relationship Reasoning (SRR)**: Select the correct object relationship displayed in the image. This is a multiple choice question with four choices. The correct choice is the ground-truth object relationship in Visual Genome dataset, and we format the question by using the object and subject name, such as "What is the relationship between girl and computer?". The three distraction choices are generated by InternVL2-40B based on the image provided with factually incorrect relationships. We also attempt to use other MLLM for this distractor generation task, including LLaVA-Next-34B and Vila-1.5, but InternVL2-40B is the one that generates the most reasonable distractors.

**Geo-Localization (GL)**: Select which county this image belongs to. We use the latitude and longitude coordinate of each image to find the corresponding county it belongs to. Only the most populated 100 counties in the United States are taken account of. We also use multiple choices format for this task type, and the distraction choices are randomly chose from the 100 counties.

**Indicator Prediction (IP)**: Predict the beautiful, wealthy and depressing level of the image from a level of 0.0 to 9.9. We use Place Pulse 2.0 dataset, which let human to make comparison between two images in multiple dimensions. Then a ranking algorithm is used to assign ground-truth labels for the images, and we ask the MLLMs to give a direct estimation in this range.

• **Cross-View Tasks:**

**Indicator Prediction (IP)**: Predict the median income, poverty ratio, total population (SafeGraph, 2024) and depression rate level (Lee, 2023) of a set of images in the same census tract or 1 kilometer map grid from a level of 0.0 to 9.9. We follow the normalization method used in GeoLLM (Manvi et al., 2023), scaling down the indicators to the range of 0.0 to 9.9, and ask the MLLMs to give a direct estimation in this range.

**Spatial Relationship Reasoning (SRR)**: Figure out which part of the satellite image does the street view image under the same area belong to. The answer should be selected from 'top-left', 'top-right', 'bottom-left', and 'bottom-right'.

### A.6.3 URBANVIEW BENCHMARK AND EVALUATION

We propose UrbanView Benchmark and construct corresponding evaluation methods. One thousand data points are sampled from our dataset for each street view, cross-view, and satellite indicator prediction task for the benchmark evaluation, while the original data size are kept for all the satellite tasks based on FIT-RS. For the Benchmark for evaluation, all metrics are in format of accuracy, except for the satellite multi scene classification task, which uses F1-score as a evaluation metric. The ground-truth labels in benchmark also remain identical with our dataset.

```
prompt ='''
You are a powerful street-view image captioner.
Please create captions describing the contents of the given
image.
The caption annotation procedure follows the principles of:
(1): Describing object attributes, including object
quantity, color, material, shape, size, and spatial
position (including absolute position in the image and
relative position between objects);
(2): The annotation process involves just describing
specific object;
(3): Instead of describing the imaginary content, only
describing the content one can determine confidently from
the image.
Do not describe the contents by itemizing them in list form.
Minimize aesthetic descriptions as much as possible;
(4): Please output less 35 words.
'''

Answer ='''
A three-story residential building with light green and
beige exterior, white trim, and multiple windows. A red SUV
and a black car parked on the street. A small white garage
and a tree in the background.
'''
```

**Captioner: InternVL2-40B**

Figure 11: Satellite image captioning for pretraining dataset.

```
prompt ='''
You are a powerful remote sensing and aerial image
captioner.
Please create SHORT captions describing the contents of the
given image.
The caption annotation procedure follows the principles of:
(1): Describing object attributes, including object
quantity, color, material, shape, size, and spatial
position (including absolute position in the image and
relative position between objects);
(2): The annotation process involves just describing the
overall scene of the image and some specific object;
(3): Instead of describing the imaginary content, only
describing the content one can determine confidently from
the image.
Do not describe the contents by itemizing them in list form.
Minimize aesthetic descriptions as much as possible;
(4): Please output within 25 words.
'''

Answer ='''
Aerial view of a suburban area with a mix of commercial
buildings, parking lots, and green spaces. A major road
curves through the scene, with a large white building near
the center.
'''
```

**Captioner: InternVL2-40B**

Figure 12: Street view image captioning for pretraining dataset.

## A.7 Task Examples

**Satellite Image-Scene Classification (SI-SC)**

```
prompt ='''
Classify the given image into the following classes.
Classes: smoke, taxiway, cooling_tower, goods_yard,
truck_parking, genset, stadium, runway, terminal, flood_dam,
foundation_pit, tower_crane, coal_yard, airplane,
storehouse, cement_concrete_pavement, car, substation, tank,
boarding_bridge, apron, unfinished_building, breakwater,
wind_mill, ground_track_field, lattice_tower, tennis_court,
ship_lock, chimney, arch_dam, ship, roundabout,
baseball_diamond. \nAnswer with all applicable classes
separated by commas.
'''
answers =
llava_next_llama3: ship
VILA1.5-40b: construction_site
InternVL2-40B: ship
cogvlm2-llama3-chat-19B: smoke, runway, …
llava_next_yi_34b: ship
llava_onevision_qwen2_7b_ov: ship
gpt-4o-2024-08-06: ship
Qwen-VL-Plus: ship, boat, water
GeoChat: ship, ship_lock
UrbanMLLM: ship
```

Groundtruth:
ship

Figure 13: Satellite image scene classification results.

**Street View Image-Scene Classification (SVI-SC)**

```
prompt ='''
Which scene category does this image fit into? Choose just
one from: 'Family buildings', 'Mixed residential and
commercial buildings', 'Commercial and office buildings',
'Industrial and manufacturing', 'Transportation and
utility', 'Public facilities and institutions', 'Open space
and outdoor recreation', 'Vacant land', 'Unknown'. Reply
with only one of the quoted options.
'''
answers =
llava_next_llama3: Transportation and utility
VILA1.5-40b: Mixed residential and commercial buildings
InternVL2-40B: 'Transportation and utility'
cogvlm2-llama3-chat-19B: Mixed residential and commercial
buildings
llava_next_yi_34b: 'Transportation and utility'
llava_onevision_qwen2_7b_ov: Transportation and utility
gpt-4o-2024-08-06: "Family buildings"
Qwen-VL-Plus: 'Mixed residential and commercial buildings'
GeoChat: 'Family buildings'
UrbanMLLM: Transportation and utility
```

Groundtruth:
Transportation and utility

Figure 14: Street view image scene classification results.

**Satellite Geo-Localization (SI-GL)**

```
prompt ='''
From the options below, which county or administrative
region is depicted in this image? Submit only the letter of
the correct choice.
A.  Ventura County, California
B.  Salt Lake County, Utah
C.  Baltimore County, Maryland
D.  Maricopa County, Arizona
Answer only with A, B, C, or D, without any additional text.
Example output: 'A'
'''
answers =
llava_next_llama3: D
VILA1.5-40b: A
InternVL2-40B: B
cogvlm2-llama3-chat-19B: A
llava_next_yi_34b: D
llava_onevision_qwen2_7b_ov: B
gpt-4o-2024-08-06: B
Qwen-VL-Plus: A
GeoChat: D
UrbanMLLM: B
```

Groundtruth:
B

Figure 15: Satellite image geo-localization results.

**Street View Geo-Localization (SVI-GL)**

```
prompt ='''
What is the correct county or equivalent administrative
region for the place shown in this picture? Respond with
the letter of the correct choice.
A.  Cook County, Illinois
B.  Allegheny County, Pennsylvania
C.  Baltimore City, Maryland
D.  Jefferson County, Alabama
Answer only with A, B, C, or D, without any additional text.
Example output: 'A'
'''
answers =
llava_next_llama3: B
VILA1.5-40b: A
InternVL2-40B: B
cogvlm2-llama3-chat-19B: B
llava_next_yi_34b: A
llava_onevision_qwen2_7b_ov: B
gpt-4o-2024-08-06: B
Qwen-VL-Plus: B
GeoChat: A
UrbanMLLM: B
```

Groundtruth:
B

Figure 16: Street view image geo-localization results.

