# OpenReview forum: "UrbanMLLM: Joint Learning of Cross-view Imagery for Urban Understanding"
_ICLR.cc/2025/Conference — Submitted to ICLR 2025_

### Official Review · Reviewer_NvHt · 2024-10-18

**Soundness:** 3
**Presentation:** 3
**Contribution:** 3
**Rating:** 6
**Confidence:** 4

**Summary:**

This paper introduces UrbanMLLM, a multimodal large language model designed to improve urban understanding by jointly leveraging satellite and street-view imagery. The authors propose a cross-view perceiver module that facilitates the integration of satellite and street-level details, addressing the limitations of relying solely on remote sensing data. In addition, the paper introduces an interleaved pre-training paradigm, where satellite and street-view images are paired with relevant textual descriptions, creating a richer context for urban understanding tasks.

The model is evaluated on a variety of tasks, including satellite image analysis, street-view analysis, and cross-view tasks, showcasing notable performance improvements over existing MLLMs. The results highlight UrbanMLLM’s ability to excel in both fine-grained perception tasks and more complex reasoning tasks, such as population density and spatial relationships. The extensive dataset and the diverse range of urban tasks provide a comprehensive testbed for the model’s capabilities. The authors also perform ablation studies to demonstrate the significance of each component, particularly the cross-view perceiver, in achieving performance gains.

Overall, the paper provides a robust framework for advancing urban understanding through multimodal fusion, and the results show clear improvements across several key metrics.

**Strengths:**

1. **Rich Dataset**: The paper makes use of an extensive dataset comprising over 2 million satellite and street view images, creating a large-scale cross-view interleaved pre-training dataset. This dataset greatly enhances the model's capacity for multimodal learning, allowing it to effectively capture the multi-level details of urban environments.

2. **Impressive Results**: UrbanMLLM demonstrates remarkable performance across several tasks, particularly in complex fine-grained tasks and reasoning tasks, such as object-level reasoning, spatial relationship reasoning, and depression rate prediction. The model significantly outperforms existing benchmarks, showing strong capabilities in urban understanding tasks.

3. **Diverse Task Coverage**: The experiments cover a wide range of tasks, including satellite imagery tasks, street view imagery tasks, and cross-view tasks. This diversity in task design validates the model's generalization and robustness, making the conclusions more compelling.

**Weaknesses:**

1. **Lack of Novelty**: While the paper provides a solid technical solution, it lacks innovation on the methodological front. The idea of combining satellite and street view images for urban understanding has been explored extensively in previous works. Although the cross-view perceiver module is introduced, its design is relatively simple, relying mainly on cross-attention mechanisms without deeper integration. Additionally, the interleaved data pre-training strategy is common and does not bring any groundbreaking technical advancement.

2. **Limited Model Comparisons**: The experimental section does not sufficiently compare the model with others in terms of **model size** and **training data** scale. The paper showcases UrbanMLLM's superior performance, but it remains unclear whether this improvement stems primarily from the large dataset or the model's architecture. A more in-depth discussion on these factors is necessary. Furthermore, the paper lacks comparisons with specific domain models, particularly in critical tasks such as **indicator prediction**. A comparison with models like **UrbanCLIP** and **UrbanVLP** would provide a more comprehensive evaluation of UrbanMLLM's performance in urban understanding tasks.

3. **Lack of Depth in Experimental Analysis**: While the experiments cover multiple task types, there is a lack of in-depth analysis on how the model performs across different subtasks. For example, the model excels in complex reasoning tasks like population density prediction, but it does not outperform certain closed-source models in simpler tasks like geographic location prediction. A more thorough discussion of the model's strengths and weaknesses across various tasks would add valuable insights into its overall performance.

**Questions:**

1. **Can you clarify how the cross-view perceiver module compares to more advanced fusion mechanisms used in similar multimodal tasks?**
   While the cross-attention mechanism is effective, it seems like a relatively simple approach. Have you considered more sophisticated alternatives for deeper integration between satellite and street-view imagery? If so, what were the reasons for opting for the current design?

2. **What impact do the model's size and dataset scale have on the performance?**
   It's not entirely clear whether the performance improvements are primarily driven by the large dataset and increased model parameters. Have you conducted any comparisons or ablation studies that analyze the performance of smaller models or less data? How would a smaller model perform under the same conditions?

3. **Why wasn't there a comparison with domain-specific models like UrbanCLIP or UrbanVLP in the indicator prediction task?**
   Given the task-specific nature of indicator prediction, models like UrbanCLIP and UrbanVLP could provide a more relevant baseline. Could you provide insights into why these comparisons were not included, and how you believe UrbanMLLM would fare against such models?

4. **Can you elaborate on how the model handles simpler tasks, such as geographic location prediction?**
   The paper mentions that the model falls slightly behind some closed-source models in certain tasks like geographic location prediction. Could you explain why this might be the case, and whether there are specific limitations in UrbanMLLM that contribute to this performance gap?

---

> ### Author Response · Authors · 2024-11-24
> **Response to Reviewer NvHt (Part 1)**
>
> **Q1: Lack of Novelty: While the paper provides a solid technical solution, it lacks innovation on the methodological front. The idea of combining satellite and street view images for urban understanding has been explored extensively in previous works. Although the cross-view perceiver module is introduced, its design is relatively simple, relying mainly on cross-attention mechanisms without deeper integration. Additionally, the interleaved data pre-training strategy is common and does not bring any groundbreaking technical advancement.**
>
> **Response:**
> **Difference between our module and previous works:** There are some works that have expored cross attention for image and text fusion, such as Flamingo [1] , BLIP2 [2], Qwen-VL [3], CogVLM [4]. However, these models are designed for bootstrapping language-image pre-training, and cannot be generalized to urban understanding tasks. For example, Flamingo uses cross-attention to extract fixed-length feature vectors from single images. BLIP2 and QwenVL uses the learnable query to get fusion features from image and text. CogVLM use visual-expert to align the image and text deeply. These methods are all designed for extracting features from a single image and can not be applied for the joint learning process involving two types of input images.
>
> In contrast, our module is designed to conduct the mutual fusion of cross-view image features and break the isolated visual knowledge in different views, which previous designs can not achieve. We also introduce a gating mechanism to adaptively fuse the features from different views, producing semantically rich cross-view image features.
>
> **Novelty of our interleaved pre-training:** Although the interleaved pre-training strategy has been adopted by some MLLMs, but the interleaved data is limited to general domains, with limited help for solving urban enderstanding tasks. Differently,  we are the first to use geo-tagged satellite and street-view images to construct a large-scale interleaved dataset in the urban domain and develop a powerful MLLM to universally solve urban understanding tasks. We believe this training corpus is valuable for the community, which has been validated to bring clear performance gain on wide urban understanding tasks.
>
> [1] Alayrac J B, Donahue J, Luc P, et al. Flamingo: a visual language model for few-shot learning[J]. Advances in neural information processing systems, 2022, 35: 23716-23736.
> [2] Li J, Li D, Savarese S, et al. Blip-2: Bootstrapping language-image pre-training with frozen image encoders and large language models[C]//International conference on machine learning. PMLR, 2023: 19730-19742.
> [3] Bai J, Bai S, Yang S, et al. Qwen-vl: A frontier large vision-language model with versatile abilities[J]. arXiv preprint arXiv:2308.12966, 2023.
> [4] Wang W, Lv Q, Yu W, et al. Cogvlm: Visual expert for pretrained language models[J]. arXiv preprint arXiv:2311.03079, 2023.

---

> > ### Comment · Reviewer_NvHt · 2024-11-26
> >
> > Thank you for the reply, but I will still maintain my score for this part.

---

> > > ### Author Response · Authors · 2024-11-28
> > > **Follow-up on rebuttal**
> > >
> > > Thank you for your positive feedback. We understand your decision to maintain your score for this part, and we appreciate the opportunity to clarify any remaining concerns. If there are any further concerns you would like us to address or other aspects you believe would strengthen the paper, we are very willing to discuss them.
> > >
> > > Thanks again for your valuable comments and suggestions！

---

> ### Author Response · Authors · 2024-11-24
> **Response to Reviewer NvHt (Part 2)**
>
> **Q2: Limited Model Comparisons: The experimental section does not sufficiently compare the model with others in terms of model size and training data scale. The paper showcases UrbanMLLM's superior performance, but it remains unclear whether this improvement stems primarily from the large dataset or the model's architecture. A more in-depth discussion on these factors is necessary. Furthermore, the paper lacks comparisons with specific domain models, particularly in critical tasks such as indicator prediction. A comparison with models like UrbanCLIP and UrbanVLP would provide a more comprehensive evaluation of UrbanMLLM's performance in urban understanding tasks.**
>
> **Response:** In Table 2-4, we have provided the results of UrbanMLLM with different model sizes, including 3B, 8B, and 13B. The results show that the overall performance of UrbanMLLM improves with the increase of model size. And we also provide the results of UrbanMLLM trained on different dataset scales, including 0.35M, 0.92M, and 1.86M images. The results show that the performance of UrbanMLLM improves a little with the increase of data scale. The reason might be that our pre-training data is still less than data size that scaling law requires.
> As for the comparison with other models, we have compared UrbanMLLM with other CLIP-based models, including RemoteCLIP and OpenAICLIP. The results are shown in Table. UrbanMLLM outperforms these CLIP-based models on all tasks, demonstrating its effectiveness in urban understanding tasks. UrbanCLIP and UrbanVLP do not provide trained model for fine-tuning, so we can't compare them on our dataset.
>
> **Results of satellite imagery-based urban understanding tasks:**
>
> | Data Size | Single | Multi. | OR | SRR | GL | Pop. | Night. |
> | --- | --- | --- | --- | --- | --- | --- | --- |
> | 0.35M | 0.899 | 0.816 | 0.822 | 0.644 | 0.921 | 0.904 | 0.741 |
> | 0.92M | 0.908 | 0.825 | 0.823 | 0.608 | 0.935 | 0.923 | 0.775 |
> | 1.86M | 0.910 | 0.825 | 0.821 | 0.577 | 0.924 | 0.898 | 0.789 |
>
> **Results of street view imagery-based urban understanding tasks:**
>
> | Data Size | SC | OR | LR | SRR | GL | BF | WE | DP |
> | --- | --- | --- | --- | --- | --- | --- | --- | --- |
> | 0.35M  | 0.840| 0.706 | 0.847 | 0.982 | 0.902 | 0.834| 0.777 | 0.746|
> |  0.92M | 0.835 | 0.703 | 0.825 | 0.970| 0.894 | 0.838 | 0.761 | 0.773 |
> | 1.86M | 0.842 | 0.703 | 0.814 | 0.974 | 0.902 | 0.841 | 0.778 | 0.746 |
>
> **Results of cross-view imagery-based urban understanding tasks:**
>
> | Data Size | Depr. rate | Med. income | Pov. rate | Pop. | SRR |
> | --- | --- | --- | --- | --- | --- |
> | 0.35M | 0.754 | 0.781 | 0.744 | 0.486 | 0.415 |
> | 0.92M | 0.701 | 0.805 | 0.750 | 0.555 | 0.418 |
> | 1.86M | 0.653 | 0.773 | 0.760 | 0.596 | 0.421 |
>
> **Performance comparison with CLIP-based models on satellite imagery-based indicator prediction tasks:**
>
> | Model | Pop | Nightlight |
> | --- | --- | --- |
> | RemoteCLIP | 0.766 | 0.688 |
> | OpenAICLIP | 0.782 | 0.708|
> | UrbanMLLM | **0.898** | **0.789** |
>
> **Performance comparison with CLIP-based models on street-view imagery-based indicator prediction tasks:**
>
> | Model | BF | WE | DP |
> | --- | --- | --- | --- |
> | RemoteCLIP | 0.767 | 0.725 | 0.666 |
> | OpenAICLIP | 0.788 | 0.561 | 0.697 |
> | UrbanMLLM | **0.841** | **0.778** | **0.746** |
>
> **Q3: Lack of Depth in Experimental Analysis: While the experiments cover multiple task types, there is a lack of in-depth analysis on how the model performs across different subtasks. For example, the model excels in complex reasoning tasks like population density prediction, but it does not outperform certain closed-source models in simpler tasks like geographic location prediction. A more thorough discussion of the model's strengths and weaknesses across various tasks would add valuable insights into its overall performance.**
>
> **Response:** In the satellite imagery understanding tasks, UrbanMLLM excels in complex reasoning tasks in all tasks. The main reason is that the most of models are not pretrained on the large-scale dataset of satellite images. However, in the street-view imagery understanding tasks, UrbanMLLM does not outperform other models in simpler tasks like geographic location prediction and scene classification. This is because geo-tagged street-view images are more likely to be involved as training corpus of existing MLLMs, enabling most general MLLMs to achieve superior performance on such tasks. However, in more complex tasks like object reasoning and indicator prediction, UrbanMLLM outperforms existing models, demonstrating its superior specialized ability. In the cross-view tasks, UrbanMLLM excels in most of the tasks, including depression rate prediction, median income prediction, and population density prediction and spatial relationship reasoning. However, it falls slightly behind some closed-source models in certain tasks like poverty rate prediction.

---

> > ### Comment · Reviewer_NvHt · 2024-11-26
> >
> > Thank you for the detailed reply, you have resolved my doubts about this part.

---

> ### Author Response · Authors · 2024-11-24
> **Response to Reviewer NvHt (Part 3)**
>
> **Q4: Can you clarify how the cross-view perceiver module compares to more advanced fusion mechanisms used in similar multimodal tasks? While the cross-attention mechanism is effective, it seems like a relatively simple approach. Have you considered more sophisticated alternatives for deeper integration between satellite and street-view imagery? If so, what were the reasons for opting for the current design?**
>
> **Response:** It's true that there's some more sophisticated fusion methods of satellite and street-view images for CV tasks. For example, SG-BEV [1] introduces a complex satellite-guided re-projection operation that transforms street-view features into satellite-view features and then fuses them with convolutional layers. However, such approach relies on depth information and high-resolution satellite images for image alignment, inefficient to be applied to large-scale data collection. Moreover, existing methods for urban understanding take very  simple fusion methods such as feature concatenation for fusion in UrbanVLP. By comparison, we adopts the more advanced  cross-attention fusion and found that it performs effectively in our framework, thus we decide to choose it as the final implementation.
>
> [1] Ye J, Luo Q, Yu J, et al. SG-BEV: Satellite-Guided BEV Fusion for Cross-View Semantic Segmentation[C]//Proceedings of the IEEE/CVF Conference on Computer Vision and Pattern Recognition. 2024: 27748-27757.
> [2] Hao, Xixuan, et al. UrbanVLP: A Multi-Granularity Vision-Language Pre-Trained Foundation Model for Urban Indicator Prediction" *arXiv preprint arXiv:2403.16831* (2024).
>
> **Q5:** **What impact do the model's size and dataset scale have on the performance?** **It's not entirely clear whether the performance improvements are primarily driven by the large dataset and increased model parameters. Have you conducted any comparisons or ablation studies that analyze the performance of smaller models or less data? How would a smaller model perform under the same conditions?**
>
> **Response:** Please refer to the response to Q2.
>
> **Q6: Why wasn't there a comparison with domain-specific models like UrbanCLIP or UrbanVLP in the indicator prediction task?Given the task-specific nature of indicator prediction, models like UrbanCLIP and UrbanVLP could provide a more relevant baseline. Could you provide insights into why these comparisons were not included, and how you believe UrbanMLLM would fare against such models?**
>
> **Response:** This is mainly because that these two models do not provide open-source codes or weights for reproducing. Please see the response to Q2 for more details.
>
> **Q7: Can you elaborate on how the model handles simpler tasks, such as geographic location prediction? The paper mentions that the model falls slightly behind some closed-source models in certain tasks like geographic location prediction. Could you explain why this might be the case, and whether there are specific limitations in UrbanMLLM that contribute to this performance gap?**
>
> **Response:** Since geo-location information is very common online, most MLLMs might have include such knowledge in their training corpus, thus can achieve superior performance on the geographic location prediction task. However, in more complex tasks like object reasoning and indicator prediction, UrbanMLLM outperforms existing models, demonstrating its more specialized urban understanding ability. A more carefully-tuned data mixture might help UrbanMLLM achieve a better balance among all urban understanding tasks.

---

### Official Review · Reviewer_Sahu · 2024-10-26

**Soundness:** 2
**Presentation:** 3
**Contribution:** 2
**Rating:** 5
**Confidence:** 4

**Summary:**

The paper introduces a novel multimodal large language model (MLLM) designed to integrate remote sensing and street-view imagery for a more comprehensive understanding of urban environments. The authors address the limitations of existing MLLMs that focus solely on remote sensing imagery. The key contributions include the development of a cross-view perceiver module within the MLLM architecture to facilitate the fusion of visual contexts, the creation of a large-scale multimodal urban imagery dataset with geotags and annotated texts, and a new pre-training paradigm based on structurally interleaved urban image-text documents. The model, named UrbanMLLM, is evaluated on a diverse set of 13 urban understanding tasks and demonstrates significant performance improvements over both open-sourced and closed-sourced MLLMs.

**Strengths:**

1. Originality: The paper presents a unique approach to urban understanding by jointly learning from remote sensing and street-view imagery, which is a novel contribution to the field of MLLMs.

2. Quality: The authors have constructed a large-scale dataset and developed a model architecture that addresses a significant gap in current MLLM capabilities. The experiments are thorough and well-designed to test the model's performance across a range of urban understanding tasks.

3. Clarity: The paper is organized with clear explanations of the methodology, experiments, and results.

**Weaknesses:**

1. Architecture-wise, I think the cross-attention mechanism in the cross-view perceiver is not a novel part. Hence, mode-side novelty is limited, although the interleaved image-text dataset is beneficial for the MLLM community.

2. In the evaluation, the baselines you chose are MLLMs, but it could be more comprehensive to compare yours with CLIP-based models.

3. You used InternVL2-40B to generate the captions for the UrbanView dataset. The dataset quality, especially the text quality, has not been verified/evaluated yet.

4. The statement from your abstract "MLLMs in urban studies are only developed focusing on remote sensing imagery" may be incorrect, unless you illustrate the comparison of the pertaining corpus proportion of different MLLMs.

5. Lack of bad case analysis.

**Questions:**

1. I think you mentioned the UrbanVLP using satellite and street-view images in Table 1, so I wonder if it is possible that the CLIP-based pertaining model can outperform the MLLM-based pertaining model.

2. Is the vision encoder the same for both satellite and street-view images?

3. It is recommended to validate/evaluate the quality of text parts from the UrbanView dataset. Or how do you refine the text?

4. You should include a bad case analysis of UrbanMLLM, so the community can know the gap of the current best model towards comprehensive urban understanding.

5. Does UrbanMLLM support multi-image as input? If so, possible to include an ablation study on the number of images?

6. It seems that the scaling law does not totally apply to UrbanMLLM across all urban tasks. We expect you to dive into the explanation of the difference among urban tasks in terms of UrbanMLLM performance.

---

> ### Author Response · Authors · 2024-11-24
> **Response to Reviewer Sahu (Part 1)**
>
> **Q1: Architecture-wise, I think the cross-attention mechanism in the cross-view perceiver is not a novel part. Hence, mode-side novelty is limited, although the interleaved image-text dataset is beneficial for the MLLM community.**
>
> **Response:**  The perceiver resampler in Flamingo [1] from DeepMind is proposed to extract fixed-length feature vectors from a single image. Its structure is similar to Q-former [2], both adopting the learnable query to sample feature vectors from the input image. It's not designed and not able to handle the challenges of modality fusion (cross-view imagery fusion in our work).
>
> In contrast, our method is designed for cross-view images to jointly learn the region embeddings. Our module is designed to enable the complementary fusion of cross-view knowledge and get semantically rich urban imagery features. We also introduce a gating mechanism to adaptively fuse the features from different views, which can effectively solve the problem of isolated visual knowledge in different views.
>
> [1] Alayrac J B, Donahue J, Luc P, et al. Flamingo: a visual language model for few-shot learning[J]. Advances in neural information processing systems, 2022, 35: 23716-23736.
> [2] Li J, Li D, Savarese S, et al. Blip-2: Bootstrapping language-image pre-training with frozen image encoders and large language models[C]//International conference on machine learning. PMLR, 2023: 19730-19742.
>
> **Q2: In the evaluation, the baselines you chose are MLLMs, but it could be more comprehensive to compare yours with CLIP-based models.**
>
> **Response:**
>
> Here we add more CLIP-based models for comparison, including RemoteCLIP and OpenAICLIP. The results have also been updated in Table 3 of the paper. From the result, UrbanMLLM outperforms these CLIP-based models on all tasks, demonstrating its effectiveness in urban understanding tasks. As for UrbanCLIP, they did not provide trained model for fine-tuning on our dataset, so we can't provide objective results of it.
>
> **Results on satellite imagery understanding tasks:**
>
> | Model | Pop | Nightlight |
> | --- | --- | --- |
> | RemoteCLIP | 0.766 | 0.688 |
> | OpenAICLIP | 0.782 | 0.708|
> | UrbanMLLM | **0.898** | **0.789** |
>
> **Results on street view imagery understanding tasks:**
>
> | Model | BF | WE | DP |
> | --- | --- | --- | --- |
> | RemoteCLIP | 0.767 | 0.725 | 0.666 |
> | OpenAICLIP | 0.788 | 0.561 | 0.697 |
> | UrbanMLLM | **0.841** | **0.778** | **0.746** |
>
> **Q3: You used InternVL2-40B to generate the captions for the UrbanView dataset. The dataset quality, especially the text quality, has not been verified/evaluated yet.**
>
> **Response:** During our dataset establishment, we have conducted manual examination of the text quality and found that it's OK to accurately describe the content. Therefore, we did not take complex operations to further improve it at that time. The superior performance of our model pre-trained on this dataset also confirms the quality of these data to some extent. To further enhance the text quality, we are now taking a Human-AI collaboration pipeline for further refine the annotated texts. Specifically, we first use two other powerful open-source MLLMs, VILA-1.5-40B and LLaVA-Next-34B to judge if the annotated text matches with the image.  If either of them thinks it is not a match, we will proceed to send it to GPT-4o for further examination. If GPT-4o still judges the image-caption pair mismatched, we will conduct manual refinement about the text. We preliminarily randomly sample 10000 images for test and observe that the original text quality is already good, where only around 0.13% images are judged as mismatched. We are working to polish all texts in the dataset via the above pipeline and will update the refined text data in our dataset.
>
> **Q4: The statement from your abstract "MLLMs in urban studies are only developed focusing on remote sensing imagery" may be incorrect, unless you illustrate the comparison of the pertaining corpus proportion of different MLLMs.**
>
> **Response:** Sorry for the confusion. Here, we want to emphasize that the MLLMs in urban studies are mainly developed focusing on addressing remote sensing imagery-based tasks. These models are developed by fine-tuning general MLLMs on remote sensing data and actually perform poorly on street-view understanding tasks (see the performance of GeoChat in Table 3). By comparison, our UrbanMLLM is pretrained on a large-scale interleaved image-text corpus involving both satellite and street-view images, which achieves superior performance on urban understanding tasks across satellite, street-view and cross-view domains.

---

> ### Author Response · Authors · 2024-11-24
> **Response to Reviewer Sahu (Part 2)**
>
> **Q5: Lack of bad case analysis.**
>
> **Response:** Thanks for your suggestion. We have added a bad case analysis in the revised paper. We show some examples of bad cases in scene classification and indicator prediction tasks. The results are shown in Figure 6-8. Firstly, in the scene classification task, the model misclassifies the image with a truck parking as a car parking. Although there are a few differences between the two classes, the more granular understanding of the urban environment is required to distinguish them. Secondly, in the indicator prediction task, as shown in Figure 7-8, the model predicts the population density of an urban area as 6.8 and the actual value is 9.9 using a satellite image. The model fails to capture detailed information with a single-view image, which makes it challenging for the model to learn from the limited dataset. For poverty rate prediction, the model gets a high score of 5.4, but the poverty rate is 2.6. It's may be due to the number of street view images in the dataset is not enough to learn the detailed information of the urban environment.
>
> **Q6: I think you mentioned the UrbanVLP using satellite and street-view images in Table 1, so I wonder if it is possible that the CLIP-based pertaining model can outperform the MLLM-based pertaining model.**
>
> **Response:** Thanks for your suggestion. UrbanVLP is not open-source thus hard to directly compare with it. We have compared UrbanMLLM with other CLIP-based models including RemoteCLIP and CLIP (OpenAI version). The results are shown in our response to Q2. UrbanMLLM outperforms these CLIP-based models on all tasks, demonstrating its effectiveness in urban understanding tasks.
>
> **Q7: Is the vision encoder the same for both satellite and street-view images?**
>
> **Response:** Yes, the vision encoder is the same for both satellite and street-view images. The previous remote sensing MLLMs like GeoChat and LHRS-Bot have shown that the general vision encoder can also be used to deal with satellite images, thus both types of image can share the same encoder. The difference lies in the subsequent processing of the features extracted by the vision encoder, where the satellite and street-view features have two different branches for later fusion.
>
> **Q8: It is recommended to validate/evaluate the quality of text parts from the UrbanView dataset. Or how do you refine the text?**
>
> **Response:** Please see the response to Q3.
>
> **Q9: You should include a bad case analysis of UrbanMLLM, so the community can know the gap of the current best model towards comprehensive urban understanding.**
>
> **Response:** Please see the response to Q5.
>
> **Q10: Does UrbanMLLM support multi-image as input? If so, possible to include an ablation study on the number of images?**
>
> **Response:** Yes, our model can support multi-image as input. We conduct an ablation study on the number of images (N = 2, 4, 6) for indicator prediction tasks. For example, 6 images mean one satellite image and five street-view images as input. The results are as follows. It can be seen that more images bring certain performance gain for the indicator prediction task.
>
> | Number of input images | Depr. rate | Med. income | Pov. rate | Pop. |
> | --- | --- | --- | --- | --- |
> | 2 | 0.724 | 0.766 | 0.696 | 0.596|
> | 4 | 0.772| 0.789| 0.689 | 0.596 |
> | 6 | 0.764 | 0.809 | 0.720 | 0.614 |
>
> **Q11: It seems that the scaling law does not totally apply to UrbanMLLM across all urban tasks. We expect you to dive into the explanation of the difference among urban tasks in terms of UrbanMLLM performance.**
>
> **Response:** As for the scaling law, we have conducted experiments on different model sizes and data scales. The results show that the performance of UrbanMLLM improves with the increase in model size, but the performance of UrbanMLLM has limited improvement with the increase of data scale. The reason might be that our pre-training data amount has not reached the required magnitude (e.g., tens-of-millions level). As for the difference performance among tasks, it may be due to the complexity of the tasks and corresponding data scale. For example, in the satellite imagery-based scene classification task, the distribution of images across categories is uneven, making the long-tail class prediction very challenging. In this case, the performance improvement requires much more data to train the model rather than purely increasing model size, so the scaling law may be not apparent.

---

> ### Comment · Reviewer_Sahu · 2024-11-26
>
> Thank you for the reply.
>
> I am concerned about [1] because it seems that the novelty lies in the functionality, rather than the architecture.
>
> [3]: "we first use two other powerful open-source MLLMs, VILA-1.5-40B and LLaVA-Next-34B to judge if the annotated text matches with the image. " What does this procedure mean? Do you use two judges for each pair (i.e. two matching scores)? Please elaborate on this. Besides, why not use gpt4o as a judge at first, if you believe it is the 'fairest' one? Overall, I think an end-to-end evaluation pipeline and quantitative metrics for text quality should be needed (an evaluation of the image captioning field may be a good reference).

---

> > ### Author Response · Authors · 2024-11-28
> > **Response to Reviewer Sahu (Part 3)**
> >
> > **Q1:** **I am concerned about [1] because it seems that the novelty lies in the functionality, rather than the architecture.**
> >
> > **Response:** Thank you for your feedback. We would like to further clarify that our proposed module also introduces architectural novelty compared to previous works. Specifically, our cross-view perceiver module is designed to enable the mutual fusion of cross-view image features through a cross-attention mutual fusion mechanism and adaptive gating operation. This approach addresses the challenge of isolated visual knowledge across different views—an issue that is difficult to solve using previous cross-attention-based architectures, such as those in Flamingo [1], BLIP2 [2], Qwen-VL [3], and CogVLM [4].
> >
> > These existing models are primarily designed for language-image pre-training and are not well-suited for tasks such as urban understanding. For instance, Flamingo uses cross-attention to extract fixed-length feature vectors from individual images, while BLIP2 and Qwen-VL employ learnable queries to fuse image and text features. CogVLM, on the other hand, utilizes a visual-expert mechanism to align image and text representations deeply. Considering that these existing modules can not be applied to tackle the cross-view imagery learning problem, thus it can not say we do not have architecture contribution. Additionally, existing advanced methods for urban understanding, such as UrbanVLP, still use simpler architectures like feature concatenation. In comparison, our architecture for cross-view image fusion is actually a novel one.
> >
> > [1] Alayrac J B, Donahue J, Luc P, et al. Flamingo: a visual language model for few-shot learning[J]. Advances in neural information processing systems, 2022, 35: 23716-23736.
> >
> > [2] Li J, Li D, Savarese S, et al. Blip-2: Bootstrapping language-image pre-training with frozen image encoders and large language models[C]//International conference on machine learning. PMLR, 2023: 19730-19742.
> >
> > [3] Bai J, Bai S, Yang S, et al. Qwen-vl: A frontier large vision-language model with versatile abilities[J]. arXiv preprint arXiv:2308.12966, 2023.
> >
> > [4] Wang W, Lv Q, Yu W, et al. Cogvlm: Visual expert for pretrained language models[J]. arXiv preprint arXiv:2311.03079, 2023.
> >
> > **Q2: "we first use two other powerful open-source MLLMs, VILA-1.5-40B and LLaVA-Next-34B to judge if the annotated text matches with the image. " What does this procedure mean? Do you use two judges for each pair (i.e. two matching scores)? Please elaborate on this. Besides, why not use gpt4o as a judge at first, if you believe it is the 'fairest' one? Overall, I think an end-to-end evaluation pipeline and quantitative metrics for text quality should be needed (an evaluation of the image captioning field may be a good reference).**
> >
> > **Response:** We would like to add more explanation about the data quality examination process:
> >
> > 1. After using InternVL2-40B to caption all the images, we further ask VILA-1.5-40B and LLaVA-Next-34B if the caption correctly reflects the visual features in the image respectively, and let them give a direct judgment - "yes" or "no". We don't calculate matching scores in this step. When either of them gives a "no", we ask GPT-4o to make a direct judgment again. If GPT-4o also says "no" to the pair, we will regenerate the caption using GPT-4o. After processing 10,000 image-caption pairs, we calculate the original version and partially regenerated version's CLIP score and found an enhancement of 0.0012%.
> > 2. We do not want to use GPT-4o as a judge at first simply because it costs too much (more than $60,000), since we have millions of image-caption pairs and the captions are long. And in future work, we want to further increase the pre-training data size, which makes it more unrealistic.
> > 3. We do add a quantitative metric CLIP-score for text quality evaluation, as introduced in Section A.6.2 of the Appendix.
> >
> > We hope this clarifies the novelty of our work and the evaluation process. If you have any further questions, please feel free to discuss. If you are satisfied with our response, please consider revising your score. Thanks again for your feedback.

---

> ### Author Response · Authors · 2024-12-02
> **Gentle Reminder of the Discussion Deadline**
>
> Dear Reviewer Sahu,
>
> Thank you again for your thoughtful review of our manuscript. As the discussion deadline approaches, we would appreciate your feedback on whether our revisions have addressed the concerns you raised. If any further clarification is needed, we are more than happy to provide additional information.
>
> If you find the revisions satisfactory, we would be grateful if you could reconsider your score.
>
> Thank you for your time and consideration.
>
> Best regards,
>
> Submission4194 authors

---

> ### Author Response · Authors · 2024-12-03
> **A kind ask for further consideration of our response**
>
> Dear Reviewer Sahu,
>
> Thank you once again for your valuable feedback during the review process. We have carefully addressed the concerns you mentioned and provided a detailed, point-by-point response in the rebuttal. The revised paper is available at https://anonymous.4open.science/r/UrbanMLLM-2F1E, and the following revisions have been made:
>
> 1. Cross-view perceiver: We have added a more detailed discussion on this topic in Lines 186-191 and Lines 219-223 of Section 3.2 "Cross-view Fusion-enhanced UrbanMLLM".
>
> 2. CLIP-based baselines: RemoteCLIP and OpenAICLIP have been included in Tables 2-3 of Section 4.2 "Results".
>
> 3. Dataset quality: A comprehensive description of the dataset has been added in Section A.6.2 "Dataset Structure and Construction", covering Lines 1022-1054.
>
> 4. Related work: Revisions have been made to the Abstract (Lines 13-14) and the Introduction (Lines 59-60) to clarify the statement of related work.
>
> 5. Bad case analysis: Updates have been provided in Section A.4.5 "Case Study", specifically in Lines 795-803.
>
> 6. Vision encoder: The introduction of the vision encoder has been included in Lines 199-200 of Section 3.2 "Cross-view Fusion-enhanced UrbanMLLM".
>
> 7. Quality validation/evaluation: A detailed description of the dataset’s structure and evaluation has been added in Section A.6.2, covering Lines 1022-1054.
>
> 8. Multi-image study: The relevant update has been incorporated into Section A.4.2 "Ablation Study", specifically in Lines 758-770.
>
> 9. Urban tasks and UrbanMLLM performance: Revisions clarifying the differences in UrbanMLLM performance across urban tasks have been made in Section A.4.1 "Experimental Analysis", Lines 746-755, and Section A.4.3 "Data Scale Study", Lines 773-783.
>
> We sincerely hope that the revisions and clarifications meet your expectations. As the discussion phase nears its end, we kindly ask if you could take a moment to review our responses. We understand the time constraints but hope that the revisions address your concerns and will be reflected in the final scoring.
>
> Thank you once again for your time and support in refining our work.
>
> Best regards,
>
> The Authors

---

### Official Review · Reviewer_t4j3 · 2024-10-28

**Soundness:** 3
**Presentation:** 3
**Contribution:** 2
**Rating:** 6
**Confidence:** 3

**Summary:**

This paper introduces UrbanMLLM, a new MLLM dedicated to urban understanding. Unlike previous works that primarily rely on remote sensing imagery alone, this approach integrates cross-view imagery through a proposed cross-view perceiver and a new pre-training paradigm. To effectively evaluate UrbanMLLM, this work also constructed the UrbanView benchmark, demonstrating notable improvements in urban understanding over both open-source and proprietary MLLMs.

**Strengths:**

1. Most designs appear technically correct with the paper being clear and practical to follow.

2. This work is well-organized, clearly outlining the need for enhanced urban understanding by incorporating street-view imagery with remote sensing imagery in the MLLM framework.

3. The proposed UrbanView benchmark is comprehensive, and the experimental results are promising, laying a solid foundation for future research in this domain.

**Weaknesses:**

Motivation：
The paper emphasizes that prior works have largely overlooked street-view images; however, this argument alone does not fully justify the motivation, especially as prior works are limited in number, and UrbanVLP has already explored this area to some extent. Furthermore, the three tasks—perception, reasoning, and prediction—do not seem to introduce substantial innovation or expansion.

Lake of Detail：
1. While the appendix includes high-quality visualizations of the dataset, more details are needed for a new benchmark, particularly on data refinement, which is crucial.

2. The paper does not discuss limitations or future work. Additionally, no accompanying code is provided to support its methodology and results.

3. For a work targeting urban understanding, providing information on data collection, refinement, and model training time would be highly beneficial.

Evaluation：
The paper introduces a new benchmark to comprehensively assess urban understanding capabilities. However, it lacks results on previous benchmarks, which, if included, would further validate the advantages of the proposed approach.

**Questions:**

Please refer to the Weaknesses. I'm willing to raise my score if my concerns are well addressed.

---

> ### Author Response · Authors · 2024-11-24
> **Response to Reviewer t4j3 (Part 1)**
>
> **Q1: The paper emphasizes that prior works have largely overlooked street-view images; however, this argument alone does not fully justify the motivation, especially as prior works are limited in number, and UrbanVLP has already explored this area to some extent. Furthermore, the three tasks—perception, reasoning, and prediction—do not seem to introduce substantial innovation or expansion.**
>
> **Response:** Yes, it's true that there are some works that have explored using both street-view images and satellite images in urban understanding tasks, such as UrbanVLP. However, these works can only deal with prediction tasks such as indicator prediction via end-to-end fine-tuning, but fail to conduct perception and reasoning tasks. By comparison, our UrbanMLLM provides a more general solution toward wide range of urban perception, reasoning and prediction tasks.
>
> We build UrbanView Benchmark with 13 different tasks including urban perception (scene classification, geo-localization), reasoning (object reasoning, spatial relationship reasoning, landmark reasoning) and prediction (indicator prediction) based on single-view or cross-view urban imagery. A relevant work CityBench [1] only involves very limited reasoning and prediction tasks including city inference, location inference, and population prediction. Another work Urbench [2] focuses on detailed cross-view tasks on different urban scenarios without prediction tasks. By comparison, Our benchmark is more comprehensive and has a wider coverage.
>
> [1] Feng J, Zhang J, Yan J, et al. CityBench: Evaluating the Capabilities of Large Language Model as World Model[J]. arXiv preprint arXiv:2406.13945, 2024.
> [2] Zhou B, Yang H, Chen D, et al. UrBench: A Comprehensive Benchmark for Evaluating Large Multimodal Models in Multi-View Urban Scenarios[J]. arXiv preprint arXiv:2408.17267, 2024.
>
> **Q2: While the appendix includes high-quality visualizations of the dataset, more details are needed for a new benchmark, particularly on data refinement, which is crucial.**
>
> **Response:**  We have updated a detailed description of the dataset and benchmark in A.6.1 Data Collection and introduce the data refinement in A.6.2 Dataset Structure and Construction section of the paper.
>
> **Q3: The paper does not discuss limitations or future work. Additionally, no accompanying code is provided to support its methodology and results.**
>
> **Response:**
>
> **Discussion of limitations and future work:** Although our model achieves state-of-the-art performance on a wide range of urban understanding tasks, there are still some limitations. Our dataset is limited to the United States, and the generalization of the model to other countries may require additional data collection and pre-training. Therefore, we plan to extend the dataset to cover more regions and improve the general applicability of the model, and explore the use of additional modalities for urban understanding tasks. Moreover, we will investigate the use of more advanced MLLM architectures for urban understanding tasks and provide more results to demonstrate the generalization of our method.
>
> The discussion has been added to the revised paper. We have also provided an anonymous link including our codes: https://anonymous.4open.science/r/UrbanMLLM-2F1E.

---

> ### Author Response · Authors · 2024-11-24
> **Response to Reviewer t4j3 (Part 2)**
>
> **Q4: For a work targeting urban understanding, providing information on data collection, refinement, and model training time would be highly beneficial.**
>
> **Response:**
> Thanks for the suggestion. We have updated a detailed description of the dataset in A.6.1 Data Collection and A.6.2 Dataset Structure and Construction section.
>
> **About data collection:** The images are primarily collected from two sources: Google Maps API for street view images, and ESRI for satellite imagery. For street view imagery collection, we randomly generate 2,000 random points in each census tract polygon and use their coordinates to query Google Maps API, returning street view image patches and real coordinates. We scrape over 2 million street view images and all satellite imagery of zoom level 15 across the United States. The ground truth of the dataset comes from a variety of sources, including US Census Bureau, Safegraph, VIIRS, and some open-source datasets.
>
> **About data refinement:** During our dataset establishment, we have conducted manual examination of the text quality and found that it's OK to accurately describe the content. Therefore, we did not take complex operations to further improve it at that time. The superior performance of our model pre-trained on this dataset also confirms the quality of these data to some extent. To further enhance the text quality, we are now taking a Human-AI collaboration pipeline for further refine the annotated texts. Specifically, we first use two other powerful open-source MLLMs, VILA-1.5-40B and LLaVA-Next-34B to judge if the annotated text matches with the image.  If either of them thinks it is not a match, we will proceed to send it to GPT-4o for further examination. If GPT-4o still judges the image-caption pair mismatched, we will conduct manual refinement about the text. We preliminarily randomly sample 10000 images for test and observe that the original text quality is already good, where only around 0.13% images are judged as mismatched. We are working to polish all texts in the dataset via the above pipeline and will update the refined text data in our dataset.
>
> We mostly use MLLMs to generate question variants in the instruct-tuning dataset, and most ground truths come from open-source data such as U.S. Census Bureau, VIIRS and WorldPop, thus the quality is reliable. For a few street-view imagery-based tasks without ground truths such as scene classification, we try various prompting methods and different MLLM architectures to generate the label with detailed human examinations to guarantee the quality.
>
> **Training Details:** As for training details, we train the 8B model on 8 NVIDIA A100 GPUs. In the pre-training stage, the model is trained for 3,200 steps with a batch size of 8. In the fine-tuning stage, the model is fine-tuned for 7,200 steps with a batch size of 16. The training time for the 8B model is approximately 8 hours for pre-training and 8 hours for fine-tuning. We have added it on the section of 4.1 EXPERIMENTAL SETUP.
>
>
> **Q5: The paper introduces a new benchmark to comprehensively assess urban understanding capabilities. However, it lacks results on previous benchmarks, which, if included, would further validate the advantages of the proposed approach.**
>
> **Response:** We  now provide the results of our model on another relevant benchmark CityBench [1]. We test the performance of our model on three tasks including CityInfer, LocInfer, and Population and use Accuracy, Accuracy@25km, and R-square as evaluation metrics. The results are as follows and it can be seen that our model outperforms SOTA models reported in their paper, demonstrating the high generalization ability of our model on urban understanding tasks.
>
> | Model | CityInfer | LocInfer | Population |
> | --- | --- | --- | --- |
> | Close-Source Model | 0.862 | 0.797 | 0.122 |
> | Open-Source Model | 0.574 | 0.555 | -0.113 |
> | UrbanMLLM-8B | 0.904 | 0.840 | 0.324 |
>
> [1] Feng J, Zhang J, Yan J, et al. CityBench: Evaluating the Capabilities of Large Language Model as World Model[J]. arXiv preprint arXiv:2406.13945, 2024.

---

> ### Author Response · Authors · 2024-11-28
> **Follow-up on rebuttal**
>
> Thank you for your careful review and suggestions. We hope that the revised version of the paper and the response meet your expectations. If you have any further questions, please let us know and we will provide detailed response to address your concerns. If you are satisfied with our response, please consider revising your score. Thanks again for your insightful comments and suggestions.

---

> ### Author Response · Authors · 2024-12-02
> **Gentle Reminder of the Discussion Deadline**
>
> Dear Reviewer t4j3,
>
> Thank you once again for your valuable review of our manuscript. As the discussion deadline approaches, we would like to confirm whether our responses have fully addressed your concerns. If any clarification is needed, we would be happy to provide further details. Additionally, if the revisions are satisfactory, we respectfully request that you reconsider your score.
>
> We appreciate your time and consideration.
>
> Best regards,
>
> Submission4194 authors

---

> ### Author Response · Authors · 2024-12-03
> **A kind ask for further consideration of our response**
>
> Dear Reviewer t4j3,
>
> Thank you once again for your valuable feedback during the review process. We have carefully addressed the concerns you mentioned, and a detailed point-by-point response is provided in the rebuttal. Additionally, we have made the following revisions to the paper (https://anonymous.4open.science/r/UrbanMLLM-2F1E/ICLR_2025_UrbanMLLM-V1201.pdf):
> (1) About data refinement, we have added a detailed description of the dataset in Section A.6.1 (Data Collection) and introduced data refinement in Section A.6.2 (Dataset Structure and Construction).
> (2) About limitations and future work, we have added a discussion on Section A.2 (Limitations and Future Work).
> (3) About training details, we have added training information in Line 321-373 of Section 4.1 (Experimental Setup).
> (4) About benchmark comparison, we have included results of our model on the CityBench benchmark in Section A.4.4 (Evaluation on CityBench) and discussed our advantage over other benchmarks in Line 1119-1126 of Section A.6.3 (UrbanView Benchmark and Evaluation).
> We sincerely hope that the revisions and clarifications we've made meet your expectations. As the discussion phase is nearly ending, we kindly ask you to take a moment to review our responses. While we understand the time constraints, we are hopeful that these revisions address your concerns and will be reflected in the final evaluation.
> We deeply appreciate your time and support in helping to refine our work.
>
> Best regards,
> Authors

---

### Official Review · Reviewer_CuDA · 2024-11-02

**Soundness:** 2
**Presentation:** 3
**Contribution:** 2
**Rating:** 5
**Confidence:** 5

**Summary:**

This paper proposes an urban research MLLM, called UrbanMLLM, which links visual information from cross-view urban images through a cross-attention mechanism and introduces a structured interleaved urban image-text pre-training framework. Meanwhile, the authors collect a large dataset of satellite view and street view images along with their geo-tagged and annotated texts to fill the lack of urban MLLM regarding street view-related data.

**Strengths:**

1. Collect and construct a dataset of satellite and street view images and their geo-tagged and annotated text, providing data support for subsequent research in the field.

2. Provides a framework for interacting with multi-view information, which to some extent alleviates the current problem of poor performance of Street View in the MLLM domain of urban research.

3. Although there are some problems in the experimental setup, after the adoption of the new dataset, a more obvious improvement in performance level has been achieved.

**Weaknesses:**

1. The data obtained by MLLM is inherently noisy and limited, and introducing it for training without special processing can result in models with a significant upper bound.

2. There does not seem to be a straightforward relationship between the direct information interaction of the perspective images and the performance of the larger model (e.g., Figure 13), and many MLLMs can produce correct answers without this mechanism. This may be because UrbanMLLM only focuses on the relationship between different perspective images of the same area and ignores the connection between the corresponding captions.

3. During the experiments, only UrbanMLLM was trained on the dataset collected by the authors, which is extremely unfair, and it is difficult to judge whether the difference in performance between different MLLMs is due to missing data or structural problems, or whether it is a problem with the pre-training process.

**Questions:**

1. The results of LHRS-Bot and SkysenseGPT are not shown in Table II and in Table III. What is the reason for this?

2. Although the authors have validated the effectiveness of the proposed dataset to some extent, the validation set in the experiments has labels, which are captions generated by MLLM. How can the accuracy of the benchmark be guaranteed?

3. The performance of the proposed dataset under other model architectures needs to be verified, and also how the dataset performs under other benchmarks after pre-training needs to be given.

4. What is the reason for the fact that in Table 7, none of the components of UrbanMLLM-8B have a significant impact on the performance, and there is even a large gap for certain tasks?

**Details Of Ethics Concerns:**

No ethics review is needed.

---

> ### Author Response · Authors · 2024-11-24
> **Response to Reviewer CuDA(Part 1)**
>
> **Q1: The data obtained by MLLM is inherently noisy and limited, and introducing it for training without special processing can result in models with a significant upper bound.**
>
> **Response:**
>
> **About the quality of text annotations of images:** During our dataset establishment, we have conducted manual examination of the text quality and found that it's OK to accurately describe the content. Therefore, we did not take complex operations to further improve it at that time. The superior performance of our model pre-trained on this dataset also confirms the quality of these data to some extent. To further enhance the text quality, we are now taking a Human-AI collaboration pipeline for further refine the annotated texts. Specifically, we first use two other powerful open-source MLLMs, VILA-1.5-40B and LLaVA-Next-34B to judge if the annotated text matches with the image.  If either of them thinks it is not a match, we will proceed to send it to GPT-4o for further examination. If GPT-4o still judges the image-caption pair mismatched, we will conduct manual refinement about the text. We preliminarily randomly sample 10000 images for test and observe that the original text quality is already good, where only around 0.13% images are judged as mismatched. We are working to polish all texts in the dataset via the above pipeline and will update the refined text data in our dataset.
>
> **About the quality of benchmark:**  We mostly use MLLMs to generate question variants in this part, and most ground truths  come from open-source data such as U.S. Census Bureau, VIIRS and WorldPop, thus the quality is reliable. For a few street-view imagery-based tasks without ground truths such as scene classification, we try various prompting methods and different MLLM architectures to generate the label with detailed human examinations to guarantee the quality.
>
> **Q2: There does not seem to be a straightforward relationship between the direct information interaction of the perspective images and the performance of the larger model (e.g., Figure 13), and many MLLMs can produce correct answers without this mechanism. This may be because UrbanMLLM only focuses on the relationship between different perspective images of the same area and ignores the connection between the corresponding captions.**
>
> **Response:** In urban understanding tasks, the model needs to understand the urban environment from both satellite and street-view images. During the interleaved pre-training process, our model can learn both textual and visual knowledge from the knowledge of another view.  Taking the street view imagery-based geo-localization task in Figure 13 as an example, our model can capture both object details in street-view images and region function knowledge from satellite images, enabling complementary learning and making it easier to accurately determine the location. Without our design, the model would be limited to extracting knowledge solely from the street view itself.

---

> ### Author Response · Authors · 2024-11-24
> **Response to Reviewer CuDA (Part 2)**
>
> **Q3: During the experiments, only UrbanMLLM was trained on the dataset collected by the authors, which is extremely unfair, and it is difficult to judge whether the difference in performance between different MLLMs is due to missing data or structural problems, or whether it is a problem with the pre-training process.**
>
> **Response:**
> In the ablation study, we have provided the result comparison of UrbanMLLM and the only fine-tuned VILA on our dataset. We now provide a more comprehensive comparison including  (1) VILA without any further training (2) VILA fine-tuned with our instruction tuning data and (3) our UrbanMLLM. The results demonstrate that UrbanMLLM outperforms VILA with fine-tuning on the same dataset, demonstrating the effectiveness of our interleaved pre-training process. Besides, the fine-tuned VILA performs better than zero-shot version, indicating the necessity of fine-tuning. We also agree that the structure of MLLMs has influence on the performance and will apply our pre-training and cross-view perceiver designs to more MLLM structures to demonstrate the generalization of our method.
>
> **Results of satellite imagery-based urban understanding tasks:**
>
> | Model                | Single | Multi. | OR    | SRR   | GL    | Pop.  | Night. |
> | -------------------- | ------ | ------ | ----- | ----- | ----- | ----- | ------ |
> | VILA-8B              | 0.629  | 0.157  | 0.619 | 0.380 | 0.589 | 0.710 | 0.455  |
> | VILA-8B (fine-tuned) | 0.888  | 0.806  | 0.818 | 0.604 | 0.903 | 0.869 | 0.713  |
> | UrbanMLLM-8B         | 0.910  | 0.825  | 0.821 | 0.577 | 0.924 | 0.898 | 0.789  |
>
> **Results of street view imagery-based urban understanding tasks:**
>
> | Model                | SC    | OR    | LR    | SRR   | GL    | BF    | WE    | DP    |
> | -------------------- | ----- | ----- | ----- | ----- | ----- | ----- | ----- | ----- |
> | VILA-8B              | 0.483 | 0.398 | 0.701 | 0.654 | 0.685 | 0.309 | 0.460 | 0.666 |
> | VILA-8B (fine-tuned) | 0.772 | 0.696 | 0.812 | 0.964 | 0.888 | 0.878 | 0.774 | 0.737 |
> | UrbanMLLM-8B         | 0.842 | 0.703 | 0.814 | 0.974 | 0.902 | 0.841 | 0.778 | 0.746 |
>
> **Results of cross-view imagery-based urban understanding tasks:**
>
> | Model                | Depr. rate | Med. income | Pov. rate | Pop.  | SRR   |
> | -------------------- | ---------- | ----------- | --------- | ----- | ----- |
> | VILA-8B              | 0.522      | 0.607       | 0.497     | 0.525 | 0.220 |
> | VILA-8B (fine-tuned) | 0.674      | 0.793       | 0.697     | 0.557 | 0.348 |
> | UrbanMLLM-8B         | 0.653      | 0.773       | 0.760     | 0.596 | 0.421 |
>
> **Q4: The results of LHRS-Bot and SkysenseGPT are not shown in Table II and in Table III. What is the reason for this?**
>
> **Response:** We have added LHRS-Bot as a baseline. The results have been updated in Table 2-3 of the paper, demonstrating that UrbanMLLM outperforms LHRS-Bot on all tasks. There is no open-source codes and trained models for SkysenseGPT thus it's hard for comparison.
>
> **Q5: Although the authors have validated the effectiveness of the proposed dataset to some extent, the validation set in the experiments has labels, which are captions generated by MLLM. How can the accuracy of the benchmark be guaranteed?**
>
> **Response:** We mostly use MLLMs to generate question variants in this part, and most ground truths  come from open-source data such as U.S. Census Bureau, VIIRS and WorldPop, thus the quality is reliable. For a few street-view imagery-based tasks without ground truths such as scene classification, we try various prompting methods and different MLLM architectures to generate the label with detailed human examinations to guarantee the quality.

---

> ### Author Response · Authors · 2024-11-24
> **Response to Reviewer CuDA (Part 3)**
>
> **Q6: The performance of the proposed dataset under other model architectures needs to be verified, and also how the dataset performs under other benchmarks after pre-training needs to be given.**
>
> **Response:**
> Thank you for your suggestion. We have added result comparison of fine-tuned VILA-8B and Qwen-2 VL (the most advanced open-sourced 7B MLLM) as follows. Our model outperforms VILA and is competitive with Qwen-2 VL (while we use much smaller pre-training data to achieve this).
>
> **Results of satellite imagery-based urban understanding tasks:**
>
> | Model                 | Single | Multi. | OR    | SRR   | GL    | Pop.  | Night. |
> | --------------------- | ------ | ------ | ----- | ----- | ----- | ----- | ------ |
> | VILA-8B (fine-tuned)  | 0.888  | 0.806  | 0.818 | 0.604 | 0.903 | 0.869 | 0.713  |
> | Qwen2-VL (fine-tuned) | 0.874  | 0.812  | 0.809 | 0.740 | 0.917 | 0.923 | 0.735  |
> | UrbanMLLM             | 0.910  | 0.825  | 0.821 | 0.577 | 0.924 | 0.898 | 0.789  |
>
> **Results of street view imagery-based urban understanding tasks:**
>
> | Model                 | SC    | OR    | LR    | SRR   | GL    | BF    | WE    | DP    |
> | --------------------- | ----- | ----- | ----- | ----- | ----- | ----- | ----- | ----- |
> | VILA-8B (fine-tuned)  | 0.772 | 0.696 | 0.812 | 0.604 | 0.964 | 0.888 | 0.878 | 0.774 |
> | Qwen2-VL (fine-tuned) | 0.823 | 0.700 | 0.846 | 0.982 | 0.917 | 0.858 | 0.780 | 0.723 |
> | UrbanMLLM             | 0.842 | 0.703 | 0.814 | 0.974 | 0.902 | 0.841 | 0.778 | 0.746 |
>
> **Results of cross-view imagery-based urban understanding tasks:**
>
> | Model                 | Depr. rate | Med. income | Pov. rate | Pop.  | SRR   |
> | --------------------- | ---------- | ----------- | --------- | ----- | ----- |
> | VILA-8B (fine-tuned)  | 0.674      | 0.793       | 0.697     | 0.557 | 0.348 |
> | Qwen2-VL (fine-tuned) | 0.833      | 0.783       | 0.745     | 0.623 | 0.371 |
> | UrbanMLLM             | 0.653      | 0.773       | 0.760     | 0.596 | 0.421 |
>
> We also provide the results of our model on another relevant benchmark CityBench [1]. We test the performance of our model on three tasks including CityInfer, LocInfer, and Population and use Accuracy, Accuracy@25km, and R-square as evaluation metrics. The results are as follows and it can be seen that our model outperforms SOTA models reported in their paper, demonstrating the high generalization ability of our model on urban understanding tasks.
>
> | Model | CityInfer | LocInfer | Population |
> | --- | --- | --- | --- |
> | SOTA closed-source model in CityBench | 0.862 | 0.797 | 0.122 |
> | SOTA open-source model in CityBench | 0.574 | 0.555 | -0.113 |
> | UrbanMLLM-8B (Ours) | **0.904** | **0.840** | **0.324** |
>
> [1] Feng J, Zhang J, Yan J, et al. CityBench: Evaluating the Capabilities of Large Language Model as World Model[J]. arXiv preprint arXiv:2406.13945, 2024.
>
> **Q7: What is the reason for the fact that in Table 7, none of the components of UrbanMLLM-8B have a significant impact on the performance, and there is even a large gap for certain tasks?**
>
> **Response:** In the ablation study, we conduct experiments to evaluate the effectiveness of the components of UrbanMLLM. The results show that the perceiver module and the interleaved pre-training strategy are effective for urban understanding tasks. By comparing w/o Perceiver and  w/o SI+SVI+Perceiver in Table 5-7, it can be observed that the performance of the model drops significantly when the perceiver module or the interleaved pre-training strategy is removed. And we found that the perceiver module and the interleaved pre-training strategy need to be combined to achieve the best performance.

---

> ### Author Response · Authors · 2024-11-28
> **Follow-up on rebuttal**
>
> Thank you for your thoughtful suggestions. We have revised the paper according to your comments. We hope that our revised version and response meet your expectations. If you have any further questions, please let us know and we will provide detailed response to address your concerns. If you are satisfied with our response, please consider revising your evaluation. Thanks again for your insightful comments and suggestions.

---

> ### Author Response · Authors · 2024-12-02
> **Gentle Reminder of the Discussion Deadline**
>
> Dear Reviewer CuDA,
>
> Thank you once again for your thoughtful review of our manuscript. With the discussion deadline approaching, we would appreciate it if you could let us know whether our responses have addressed your concerns. If further clarification is needed, we are happy to provide additional information. If the revisions meet your expectations, we kindly ask that you reconsider the paper’s score.
>
> Thank you for your time and feedback.
>
> Best regards,
>
> Submission4194 authors

---

> > ### Comment · Reviewer_CuDA · 2024-12-02
> > **The author's rebuttal does not address my concerns well**
> >
> > Thanks to the authors for their efforts and serious responses to the rebuttal, some of which explained my confusion well, but there are still some problems:
> >
> > Regarding the quality part of the image-text annotation, the human-computer collaboration pipeline is not visible at all in the paper and should be emphasized if manual work is introduced in the generation process, so as not to bring a wrong perception to the readers or subsequent researchers, and confuse the difference between the pure MLLM and the hybrid human-computer pipeline. The authors do not seem to indicate that this issue will be further discussed and corrected in subsequent editions. In the meantime, this work introduces several existing and unfinished MLLMs such as GPT4 to assist in the screening process, and according to the authors' given motivations and experimental results, these MLLMs perform relatively poorly in the urban task, which makes their mismatch rates unconvincing. For this problem, the solution strategy for future improvement and the description of the problem itself did not convince me.
> >
> > For Q3, the corresponding experimental description somewhat alleviated my doubts about the validity of the architecture proposed in this paper. However, as the authors mentioned, "more MLlM structures to demonstrate the generalization of our method“, this question is still not fully answered because only a single base model is used.
> >
> > The author's answer to Q4 confuses me as to why LHRs-Bot was chosen as the baseline, whether it means that it has the worst performance compared to the other comparative algorithms, or that it did not improve the urban task accordingly.
> >
> > The answer in Q7 does not directly answer my question, which focuses on the drawbacks or limitations of the proposed method for large models or models pre-trained with larger data. The comparison with Owen-2 vl in Q6 also happens to illustrate this point.

---

> > > ### Author Response · Authors · 2024-12-03
> > > **Response to Reviewer CuDA (Part 4)**
> > >
> > > **Q1: Regarding the quality part of the image-text annotation, the human-computer collaboration pipeline is not visible at all in the paper and should be emphasized if manual work is introduced in the generation process, so as not to bring a wrong perception to the readers or subsequent researchers, and confuse the difference between the pure MLLM and the hybrid human-computer pipeline. The authors do not seem to indicate that this issue will be further discussed and corrected in subsequent editions. In the meantime, this work introduces several existing and unfinished MLLMs such as GPT4 to assist in the screening process, and according to the authors' given motivations and experimental results, these MLLMs perform relatively poorly in the urban task, which makes their mismatch rates unconvincing. For this problem, the solution strategy for future improvement and the description of the problem itself did not convince me.**
> > >
> > > **Response:**
> > > Thank you for your feedback.
> > >
> > > **About Human-Computer Collaboration Process:**
> > > We've added a detailed description of the human-computer collaboration process in Section A.6.2 "Dataset Structure and Construction" in the appendix. This clarifies the role of human annotators in the data generation and validation pipeline, complemented by MLLMs for scaling and refinement.
> > >
> > > **About Use of GPT-4 and Other MLLMs:**
> > > GPT-4 and other MLLMs are primarily used for text annotation, not question answering. This approach is consistent with studies in remote sensing (e.g., [1], [2]). Since MLLMs are not specifically tuned for urban tasks, we’ve developed a large-scale instruct-tuning dataset to fine-tune them for urban understanding.
> > > [1] Pang C, Wu J, Li J, et al. H2RSVLM: Towards Helpful and Honest Remote Sensing Large Vision Language Model[J]. arXiv preprint arXiv:2403.20213, 2024.
> > >
> > > [2] Hao, Xixuan, et al. UrbanVLP: A Multi-Granularity Vision-Language Pre-Trained Foundation Model for Urban Indicator Prediction" *arXiv preprint arXiv:2403.16831* (2024).
> > >
> > > **Q2: For Q3, the corresponding experimental description somewhat alleviated my doubts about the validity of the architecture proposed in this paper. However, as the authors mentioned, "more MLlM structures to demonstrate the generalization of our method", this question is still not fully answered because only a single base model is used.**
> > >
> > > **Response:**
> > > Thank you for your feedback. In our experiments, we used a single base model to demonstrate the effectiveness of our proposed architecture. However, we also tested the structure with models of different sizes to validate its performance. Due to time and resource limitations, we plan to investigate more MLLM structures in future work to further assess and demonstrate the generalization ability of our approach.

---

> > > ### Author Response · Authors · 2024-12-03
> > > **Response to Reviewer CuDA (Part 5)**
> > >
> > > **Q3: The author's answer to Q4 confuses me as to why LHRs-Bot was chosen as the baseline, whether it means that it has the worst performance compared to the other comparative algorithms, or that it did not improve the urban task accordingly.**
> > >
> > > **Response:**
> > > Thank you for your feedback. As mentioned in your previous review "The results of LHRS-Bot and SkysenseGPT are not shown in Table II and Table III." We followed your suggestion and have now added the results of LHRS-Bot, as it is a state-of-the-art model for remote sensing tasks and is open-sourced, ensuring a fair and transparent comparison.
> > >
> > >
> > >
> > >
> > > **Q4: The answer in Q7 does not directly answer my question, which focuses on the drawbacks or limitations of the proposed method for large models or models pre-trained with larger data. The comparison with Qwen-2 vl in Q6 also happens to illustrate this point.**
> > >
> > > **Response:** Thank you for your feedback.
> > >
> > > In our ablation study, we compared different components of our method under the same experimental settings. As shown in Tables 5-7, removing the Perceiver module or the interleaved pre-training strategy significantly reduces the model's performance. This indicates that combining the Perceiver module with the interleaved pre-training strategy yields the best results.
> > >
> > > Moreover, we have conducted experiments with varying model sizes, as presented in Tables 2-4, and observed that performance generally improves as the model size increases. Additionally, we evaluated the effect of different pre-training data sizes (see Appendix Tables 9-11) and find that the performance improves slightly with larger datasets. We think it might be because our pre-training data is still smaller than the scale required to fully leverage the scaling laws for large models.
> > >
> > > Our method is overall competitive with Qwen-2 VL, while it is pre-trained on much larger datasets than us, so our method can serve as a balance between accuracy and computation costs. What's more, our method is not orthogonal to more powerful base LLMs like Qwen-2 VL, and we plan to apply our method to more LLM architectures and add these results in the future version of the paper.
> > >
> > > Thanks again for your feedback, which has helped us clarify these points in the revised manuscript. If our responses satisfactorily address your concerns, we would appreciate it if you could consider your score for this review.

---

> ### Author Response · Authors · 2024-12-04
> **A kind ask for further consideration of our response**
>
> Dear Reviewer CuDA,
>
> Thanks again for your valuable and continued feedback during the review process, which is very constructive to improve this work. We have carefully addressed the concerns you mentioned, and provided a detailed point-by-point response during the discussion phase. In addition, we have made the corresponding revisions in the paper (please see: https://anonymous.4open.science/r/UrbanMLLM-2F1E/ICLR_2025_UrbanMLLM-V1201.pdf), which are summarized as follows:
>
> (1) About more details especially quality control of our dataset and benchmark, we have added a detailed description of the dataset in Section A.6.1 (Data Collection) and introduced data refinement in Section A.6.2 (Dataset Structure and Construction).
>
> (2) About the experimental comparison with other trained models, we have updated results in Table 2-4 and added more discussion in Section 4.2.
>
> (3) About results on other benchmarks, we have included results of our model on the CityBench benchmark in Section A.4.4 (Evaluation on CityBench) and discussed our advantage over other benchmarks in Line 1119-1126 of Section A.6.3 (UrbanView Benchmark and Evaluation).
>
> (4) About the limitation of our method for large models or models pre-trained with larger data, we have added more discussion and results in Section A.4.1 and Section A.4.3 of the Appendix.
>
> We sincerely hope that these revisions align with your expectations. As the discussion phase is nearly ended, we kindly ask you to take a moment to further consider our responses. While we understand the time constraints, we hope that these revisions address your concerns and will be reflected in the final scoring. Thank you again for your valuable time and effort in refining our work.
>
> Best regards,
>
> Authors

---

### Official Review · Reviewer_iqRR · 2024-11-10

**Soundness:** 2
**Presentation:** 3
**Contribution:** 3
**Rating:** 6
**Confidence:** 4

**Summary:**

The authors created a multimedia LLM (MLLM) with the purpose of solving urban
understanding tasks. Specifically, the authors trained their UrbanMLLM model
with satellite and street-view images, in addition to text. The UrbanMLLM
includes a cross-view perceiver module to integrate satellite and cross-view
images. A pre-training method is used to then integrate the text.

**Strengths:**

S1: The authors collected a dataset that includes both satellite and cross-view
images. Though, not much details are presented about this dataset.

S2: The UrbanMLLM model performs well against general MLLM models.

S3: The authors tested a large variety of urban understanding tasks.

**Weaknesses:**

W1: Other than stating that 2 million satellite and cross-view images were
used, not much information is presented about the dataset. For example, where
did the ground truth information to calculate accuracy come from? Also, the
authors state that the dataset covers the whole of the US. I am sure that,
just for street-view images, there are more than 2 million in the US. How
did the authors select the used images and how uniform or concentrated did
they select images for certain cities?

W2: The utilized perceiver module in the UrbanMLLM is a previously proposed
component by DeepMind, hence the novelty of the architecture seems a bit
limited.

W3: It seems the performance of UrbanMLLM was compared against general MLLMs
which were not trained on the same dataset. Since only UrbanMLLM was trained on
both the satellite and cross-view images from the authors dataset it is not
very surprising that UrbanLLM's performance is overall better. Thus, the
comparison in this paper does not seem to be quite apples-to-apples.

**Questions:**

Please see my comments under Weaknesses. I would be interested in the authors'
explanations on the issues that I listed there.

Page 1:
In the abstract and introduction, I think it would be good if the authors
could be a bit more specific about what kind of applications their UrbanMLLM
is supposed to support. There are many urban applications and clearly UrbanMLLM
is only suitable and targeted towards a subset of them.

Page 2:
The authors state that the dataset covers the whole United States. They also
mention that 2 million images where used. I would assume that coverage of the
whole US would require more images. So this dataset must be very sparse. How
did the authors decide which images to select?

Page 2:
For some urban computing tasks there exist specialized architectures, e.g.,
for cross-view localization. The authors may want to refer to some of the
work, and also make it clear that they have a more general model in mind.

Page 5:
"imapact" -> impact

Page 7:
"The results showcase that UrbanMLLM achieves state-of-the-art performance,
which successfully ."
There is some text missing here at the end of this sentence.

---

> ### Author Response · Authors · 2024-11-24
> **Response to Reviewer iqRR (Part 1)**
>
> **Q1: Other than stating that 2 million satellite and cross-view images were used, not much information is presented about the dataset. For example, where did the ground truth information to calculate accuracy come from? Also, the authors state that the dataset covers the whole of the US. I am sure that, just for street-view images, there are more than 2 million in the US. How did the authors select the used images and how uniform or concentrated did they select images for certain cities?**
>
> **Response:** We have updated a detailed description of the dataset in A.6.2 Dataset Structure and Constrcution section.
>
> **About the source of ground truths:** The ground truth of the dataset comes from a variety of sources, including US Census Bureau, Safegraph, VIIRS, and a lot of open-source datasets. We do normalization for numerical indicator ground truth and reorganize the format of these ground truth data to build the instruct-tuning dataset.
>
> **About the coverage:** The street view imagery cover 71,433 out of all 73,868 census tracts in the United States (96.7% coverage), and each satellite imagery is matched with street view imagery using a 1km $\times$ 1km grid.
>
> **About the imagery selection:** The street view imagery amount in U.S. is indeed much larger than 2 million, however, due to the coordinate query method of Google API, we can only randomly generate thousands of points in each census tract polygon and use these points to query street view images, so it is not quite possible to collect all the street view images in a census tract considering the massive amount of time it requires. In fact, in some less populated areas, it is quite hard to get street view images because the randomly generated query coordinate points in these areas are always off-road, which is also the main reason for the missing 3.3% coverage of census tracts. In the end, we chose to randomly select about 200 images in each census tract, which we reckon is dense enough to make a good representation of the region according to previous works such as Urban2Vec [1] and Devil in the Landscapes [2], which both use around 40 images to represent communities with a similar size.
>
> **About the uniformity of images for certain cities:** The data distribution is largely conform to the distribution of population density in the United States. Census tracts in the United States are primarily divided based on population size, and in each census tract polygon we randomly sample similar size of images. In terms of cities, larger cities usually have more census tracts, therefore more images will be used in these cities. However, as mentioned above, we still have a very good coverage of the whole country because we sample all the census tracts.
>
> [1] Wang, Zhecheng, Haoyuan Li, and Ram Rajagopal. "Urban2vec: Incorporating street view imagery and pois for multi-modal urban neighborhood embedding." Proceedings of the AAAI Conference on Artificial Intelligence. Vol. 34. No. 01. 2020.
> [2] Han, Zhenyu, et al. "Devil in the Landscapes: Inferring Epidemic Exposure Risks from Street View Imagery." Proceedings of the 31st ACM International Conference on Advances in Geographic Information Systems. 2023.
>
>
>
> **Q2: The utilized perceiver module in the UrbanMLLM is a previously proposed component by DeepMind, hence the novelty of the architecture seems a bit limited.**
>
> **Response:**
>
> The perceiver resampler in Flamingo [1] from DeepMind is proposed to extract fixed-length feature vectors from a single image. Its structure is similar to Q-former [2], both adopting the learnable query to sample feature vectors from the input image. It's not designed and not able to handle the challenges of modality fusion (cross-view imagery fusion in our work).
>
> In contrast, our method is designed for cross-view images to jointly learn the region embeddings. Our module is designed to enable the complementary fusion of cross-view knowledge and get semantically rich urban imagery features. We also introduce a gating mechanism to adaptively fuse the features from different views, which can effectively solve the problem of isolated visual knowledge in different views.
>
>
> [1] Alayrac J B, Donahue J, Luc P, et al. Flamingo: a visual language model for few-shot learning[J]. Advances in neural information processing systems, 2022, 35: 23716-23736.
> [2] Li J, Li D, Savarese S, et al. Blip-2: Bootstrapping language-image pre-training with frozen image encoders and large language models[C]//International conference on machine learning. PMLR, 2023: 19730-19742.

---

> > ### Author Response · Authors · 2024-11-28
> > **Follow-up on rebuttal**
> >
> > We have updated the manuscript based on your comments. We hope the revised version could meet your expectations. If you have any further questions, please let us know and we will provide detailed response to address your concerns. If you are satisfied with the response and, we kindly ask you to consider revising your score. Thanks again for your insightful comments and suggestions.

---

> > ### Comment · Reviewer_iqRR · 2024-11-29
> >
> > I appreciate the authors's response to my comments. Some of my questions have been answered. For example, it is good that the authors have included more details about the dataset in the appendix of the paper. However, a few items are still not very clear to me.
> >
> > Re: Q2. In terms of the perceiver module, it seems the original perceiver architecture was proposed by DeepMind in [3], which is not the paper that was quoted in the response.
> >
> > [3] Jaegle, Andrew; Gimeno, Felix; Brock, Andrew; Zisserman, Andrew; Vinyals, Oriol; Carreira, Joao (2021-06-22). "Perceiver: General Perception with Iterative Attention". arXiv:2103.03206.
> >
> > I think the authors should cite the original work and outline the differences of their own perceiver component to this original one in the paper.

---

> > > ### Comment · Reviewer_iqRR · 2024-11-29
> > >
> > > Re: Q3. Thank you for the more detailed ablation study. I have looked at the revised PDF after the rebuttal and I don't seem to find the table in your  response above in the paper. Does the after-rebuttal version not include the full ablation study?

---

> > > > ### Author Response · Authors · 2024-11-29
> > > > **Response to Reviewer iqRR (Part 4)**
> > > >
> > > > **Q1:** **Re: Q2. In terms of the perceiver module, it seems the original perceiver architecture was proposed by DeepMind in [3], which is not the paper that was quoted in the response. I think the authors should cite the original work and outline the differences of their own perceiver component to this original one in the paper.**
> > > >
> > > > **Response:**
> > > > We thank the reviewer for pointing out the missing citation. We will include the reference to the original Perceiver paper in the final version of the manuscript.
> > > > In the Perceiver paper [3], the authors did not focus on fusing features from different modalities. Instead, they use a unified Transformer architecture for multiple modalities and employ a low-dimensional learnable query to learn high-dimensional features.
> > > >
> > > > In contrast, our Cross-View Perceiver module is specifically designed to capture the complementary relationships between different views of the input images, allowing them to mutually enhance each other's features. This is a key distinction from the original Perceiver. Furthermore, we introduce a gating mechanism that adaptively fuses features from different views, enabling Multimodal Large Language Models (MLLMs) to more effectively integrate and leverage information from diverse sources.
> > > >
> > > > [3] Jaegle, Andrew; Gimeno, Felix; Brock, Andrew; Zisserman, Andrew; Vinyals, Oriol; Carreira, Joao (2021-06-22). "Perceiver: General Perception with Iterative Attention". arXiv:2103.03206.
> > > >
> > > > **Q2:** **Re: Q3. Thank you for the more detailed ablation study. I have looked at the revised PDF after the rebuttal and I don't seem to find the table in your response above in the paper. Does the after-rebuttal version not include the full ablation study?**
> > > >
> > > > **Response:** We apologize for the confusion. In the "Response to Reviewer iqRR (Part 2)" table, "VILA-8B (fine-tuned)" refers to the same model as "w/o SI+SVI+Perceiver" in the paper. Since the performance of VILA-40B has already been included in Tables 3-5, we omitted it from the ablation study. We will add the VILA-8B results to the ablation study in the final version of the paper.
> > > >
> > > > Thanks again for your insightful comments and suggestions.

---

> > > > > ### Comment · Reviewer_iqRR · 2024-11-30
> > > > >
> > > > > Thank you for your response. Just to clarify, we - the reviewers - are interested in how you will revise your paper. The comments that we all make here in this dialogue will eventually disappear, only your paper will remain. That is why we point out certain issues and would like to hear from you how you will address them in the paper.
> > > > > You did this in your response above, which is what I was looking for and appreciate. Btw, make sure that the terminology is clear and consistent. That "VILA-8B (fine-tuned)" is the same as "w/o SI+SVI+Perceiver" would likely not occur to most readers.

---

> > > > > > ### Author Response · Authors · 2024-11-30
> > > > > > **Response to Reviewer iqRR (Part 5)**
> > > > > >
> > > > > > **Q: Just to clarify, we - the reviewers - are interested in how you will revise your paper. The comments that we all make here in this dialogue will eventually disappear, only your paper will remain. That is why we point out certain issues and would like to hear from you how you will address them in the paper. You did this in your response above, which is what I was looking for and appreciate. Btw, make sure that the terminology is clear and consistent. That "VILA-8B (fine-tuned)" is the same as "w/o SI+SVI+Perceiver" would likely not occur to most readers.**
> > > > > >
> > > > > > **Response:**
> > > > > > Thank you for your further comment. We have updated the relevant sections of the paper to ensure clear and consistent terminology. The revised pdf file has been updated in https://anonymous.4open.science/r/UrbanMLLM-2F1E, including the following revisions:
> > > > > >
> > > > > > - **Q1** (Response to Reviewer iqRR, Part 1): A detailed description of the dataset has been added in Section A.6.1 (Dataset Collection) and Section A.6.2 (Dataset Structure and Construction), covering pages P18-P21.
> > > > > >
> > > > > > - **Q2** (Response to Reviewer iqRR, Part 1): The relevant update has been incorporated in Section 3.2 (Cross-View Fusion-Enhanced UrbanMLLM), specifically in lines 219-223.
> > > > > >
> > > > > > - **Q3** (Response to Reviewer iqRR, Part 2): Revisions have been made in Section 4.2 (Results), including lines 441-444, lines 480-482, and Tables 3-5.
> > > > > >
> > > > > > - **Q4** (Response to Reviewer iqRR, Part 2): The Introduction section has been updated in lines 92-95.
> > > > > >
> > > > > > - **Q5** (Response to Reviewer iqRR, Part 2): Updates are provided in Section A.6.1 (Data Collection), covering lines 955-999.
> > > > > >
> > > > > > - **Q6** (Response to Reviewer iqRR, Part 2): The relevant update is reflected in Section 3.2 (Cross-View Fusion-Enhanced UrbanMLLM), in lines L157-L160.
> > > > > >
> > > > > > - **Q7** (Response to Reviewer iqRR, Part 3): Corrections have been implemented in the Experiments section, in line 255.
> > > > > >
> > > > > > - **Q8** (Response to Reviewer iqRR, Part 3): Revisions have been made in the Experiments section, specifically in lines 421-424.
> > > > > >
> > > > > >
> > > > > > All of the revised parts are highlighted in red. We hope these revisions address your concerns and improve the clarity and consistency of the paper. If you have any further questions or concerns, please feel free to let us know. We appreciate your feedback and suggestions, which have significantly contributed to enhancing the quality of our work. If our response and revision aligns with your expectation, we kindly request that you reconsider the score. Thanks for your valuable suggestions again!

---

> > > > > > > ### Author Response · Authors · 2024-12-03
> > > > > > >
> > > > > > > Thank you for your thoughtful feedback on our work and for raising the score from 3 to 6. We are glad that our revisions address your concerns. Your constructive feedback has been invaluable in enhancing the quality of our paper.
> > > > > > >
> > > > > > > Best Regards,
> > > > > > >
> > > > > > > Authors

---

> ### Author Response · Authors · 2024-11-24
> **Response to Reviewer iqRR (Part 2)**
>
> **Q3: It seems the performance of UrbanMLLM was compared against general MLLMs which were not trained on the same dataset. Since only UrbanMLLM was trained on both the satellite and cross-view images from the authors dataset it is not very surprising that UrbanLLM's performance is overall better. Thus, the comparison in this paper does not seem to be quite apples-to-apples.**
>
> **Response:**
>
> In the ablation study of the paper, we have provided the results of the fine-tuned version of VILA with our dataset, which has lower performance than our method. We now provide comparisons with (1) VILA without any further training (2) VILA fine-tuned with our instruction tuning data and (3) our UrbanMLLM as follows. The results demonstrate that UrbanMLLM outperforms VILA with fine-tuning on the same dataset, demonstrating the effectiveness of our method.
>
> **Results of satellite imagery-based urban understanding tasks:**
>
> | Model | Single | Multi. | OR | SRR | GL | Pop. | Night. |
> | --- | --- | --- | --- | --- | --- | --- | --- |
> | VILA-8B| 0.629 | 0.157 | 0.619 | 0.380 | 0.589 | 0.710 | 0.455 |
> | VILA-8B (fine-tuned) | 0.888 | 0.806 | 0.818 | 0.604 | 0.903 | 0.869 | 0.713 |
> | UrbanMLLM-8B | 0.910 | 0.825 | 0.821 | 0.577 | 0.924 | 0.898 | 0.789 |
>
> **Results of street view imagery-based urban understanding tasks:**
>
> | Model | SC | OR | LR | SRR | GL | BF | WE | DP |
> | --- | --- | --- | --- | --- | --- | --- | --- | --- |
> | VILA-8B | 0.483 | 0.398 | 0.701 | 0.654 | 0.685 | 0.309 | 0.460 | 0.666 |
> | VILA-8B (fine-tuned) | 0.772 | 0.696 | 0.812 | 0.964 | 0.888 | 0.878 | 0.774 | 0.737 |
> | UrbanMLLM-8B | 0.842 | 0.703 | 0.814 | 0.974 | 0.902 | 0.841 | 0.778 | 0.746 |
>
> **Results of cross-view imagery-based urban understanding tasks:**
>
> | Model | Depr. rate | Med. income | Pov. rate | Pop. | SRR |
> | --- | --- | --- | --- | --- | --- |
> | VILA-8B | 0.522 | 0.607 | 0.497 | 0.525 | 0.220 |
> | VILA-8B (fine-tuned) | 0.674 | 0.793 | 0.697 | 0.557 | 0.348 |
> | UrbanMLLM-8B | 0.653 | 0.773 | 0.760 | 0.596 | 0.421 |
>
>
>
> **Q4: In the abstract and introduction, I think it would be good if the authors could be a bit more specific about what kind of applications their UrbanMLLM is supposed to support. There are many urban applications and clearly UrbanMLLM is only suitable and targeted towards a subset of them.**
>
> **Response:**
> Thanks for the suggestion. We have added more details about the applications that UrbanMLLM can support in Introduction.
>
> In summary, our model is focused on urban understanding tasks using satellite and street-view images including: scene classification, object reasoning, spatial relationship reasoning, geo-localization, indicator prediction and landmark reasoning.
>
>
> **Q5: The authors state that the dataset covers the whole United States. They also mention that 2 million images where used. I would assume that coverage of the whole US would require more images. So this dataset must be very sparse. How did the authors decide which images to select?**
>
> **Response:** As mentioned in Q1, we randomly sample thousands of points in each census tract polygon and use these points to query street view images. About 200 random but uniformly distributed images in each census tract are selected, we observe that these images have been dense enough to serve as a good representation of the region. Moreover, we notice that there are only around 36 images used in previous works to represent communities with similar size, which can already produce good results in indicator prediction tasks. Therefore, we believe the sampled images in our dataset are well representative.
>
> **Q6: For some urban computing tasks there exist specialized architectures, e.g., for cross-view localization. The authors may want to refer to some of the work, and also make it clear that they have a more general model in mind.**
>
> **Response:** As for other urban computing tasks using satellite imagery and street-view image like cross-view localization, there are indeed some specialized models, such as Sample4Geo [1] and MFRGN [2]. However, these models are designed for specific tasks and can not be generalized to other urban understanding tasks. By comparison, our model is more general and can be applied to a wide range of urban understanding tasks, including scene classification, object reasoning, spatial relationship reasoning, geo-localization, indicator prediction, landmark reasoning, etc.
>
> [1] Deuser F, Habel K, Oswald N. Sample4geo: Hard negative sampling for cross-view geo-localisation[C]//Proceedings of the IEEE/CVF International Conference on Computer Vision. 2023: 16847-16856.
> [2] Wang Y, Zhang J, Wei R, et al. MFRGN: Multi-scale Feature Representation Generalization Network for Ground-to-Aerial Geo-localization[C]//Proceedings of the 32nd ACM International Conference on Multimedia. 2024: 2574-2583.

---

> ### Author Response · Authors · 2024-11-24
> **Response to Reviewer iqRR (Part 3)**
>
> **Q7: "imapact" -> impact**
>
> **Response:** Thanks for pointing out, we have corrected it.
>
> **Q8: "The results showcase that UrbanMLLM achieves state-of-the-art performance, which successfully ." There is some text missing here at the end of this sentence.**
>
> **Response:** Thanks for the pointing out and we have revised the paper to correct that. This revised sentence is "The results showcase that UrbanMLLM achieves state-of-the-art performance, which successfully demonstrates the effectiveness of the proposed model for urban understanding tasks".

---

> ### Author Response · Authors · 2024-12-02
> **Gentle Reminder of the Discussion Deadline**
>
> Dear Reviewer iqRR,
>
> Thank you again for reviewing our manuscript. As the discussion deadline approaches, we’d like to confirm whether our responses have addressed your concerns. If any clarification is needed, we are happy to provide further details. Additionally, if you feel the revisions are satisfactory, we would appreciate it if you could reconsider the paper’s score.
>
> Thank you for your time and consideration.
>
> Best regards,
>
> Submission4194 authors

---

### Author Response · Authors · 2024-12-04
**A General Response from Authors**

We deeply appreciate the valuable and constructive comments provided by all the reviewers, along with their detailed suggestions that have greatly enhanced the quality of our work. We are also grateful for the reviewers' active feedback throughout the discussion phase and for acknowledging our responses and revisions to the paper (e.g., "serious responses to the rebuttal, some of which explained my confusion well" mentioned by Reviewer CuDA and "you have resolved my doubts about this part" mentioned by Reviewer NvHt). Moreover, Reviewer iqRR has **raised the score from 3 to 6, indicating the acknowledgment of our response and the improvement of the paper**. Thanks for all of your effort in reviewing and enhancing our work.

---------------------

We are grateful for the received positive feedback from reviewers including:

**About the novelty of our proposed UrbanMLLM:**

The model "to some extent **alleviates the current problem of poor performance of Street View in the MLLM domain** of urban research" (Reviewer CuDA). The work presents "**a** **unique approach** to urban understanding by jointly learning from remote sensing and street-view imagery, which is **a novel contribution to the field of MLLMs**" (Reviewer Sahu).

**About the impressive performance and rich evaluation of our method:**

The model "**performs well against** general MLLM models" and the work "tested a large variety of urban understanding tasks" (Reviewer iqRR).  "After the adoption of the new dataset, a **more obvious improvement** in performance level has been achieved"  (Reviewer CuDA). The experimental results are "promising, **laying a solid foundation for future research in this domain**" (Reviewer t4j3). The model "**addresses a significant gap** in current MLLM capabilities. The experiments are **thorough and well-designed**" (Reviewer Sahu). The model "demonstrates **remarkable performance** across several tasks" and "**significantly outperforms** existing benchmarks, showing **strong capabilities** in urban understanding tasks" (Reviewer NvHt).

**About the contribution of our proposed dataset and benchmark:**

The constructed dataset and benchmark are "**comprehensive and laying a solid foundation for future research**" (Reviewer t4j3). The dataset can "**support for subsequent research in the field**" (Reviewer CuDA). This dataset "**greatly enhances the model's capacity** for multimodal learning, allowing it to effectively capture the multi-level details of urban environments"  (Reviewer NvHt).

**About the paper writing:**

This work is "**well-organized, clearly outlining**...." and "**clear, practical to follow**" (Reviewer t4j3).  The paper is "**organized with clear explanations** of the methodology, experiments, and results" (Reviewer Sahu).

---------------------

During the discussion phase, we have provided detailed point-to-point responses to all reviewers' concerns and have updated these revisions in the paper, which are briefly summarized as follows:

- **About the details of our dataset and benchmark:** We have added a detailed description of the dataset and introduced our data quality control approach in Section A.6. We promise to make the dataset and benchmark with our model codes fully open-source to benefit the community.
- **About more experimental comparison:** We have supplemented experimental results of other trained MLLMs (including VILA-8B, Qwen2-VL), CLIP-based models (including CLIP, RemoteCLIP), and remote sensing MLLMs (LHRS-Bot) in Table 2-4. We  have also added an analysis of these results in Section 4.2. We have added more ablation studies about data scale and input image number in Section A.4.2 and A.4.3. We have added the results of our model on other benchmarks (CityBench) in Section A.4.4.
- **About our method contribution:** We have added more discussion of our technical contribution in the model design and the key difference with previous works (e.g., Deepmind's perceiver, previous cross-view image fusion methods) in Section 3.2.
- **About the clear clarification of applications our model can support:** We have clarified what tasks and applications UrbanMLLM can support in the Introduction.
- **About more precise descriptions of previous works:** We have refined the description of previous works using both satellite and street-view images for urban understanding (e.g., UrbanVLP) and our main difference in Introduction and Section 2.2.
- **About more discussion of the limitation:** We have added a discussion of the limitation of the work in Section A.2 and bad case analysis in Section A.4.5.

Thanks again for all the reviewers' time and effort in enhancing our work and we believe the current paper has achieved a significant improvement with your valuable suggestions.

Best regards,

Authors

---

### Meta-Review · Area_Chair_c9cw · 2024-12-19

**Metareview:**

This paper proposed a multimodal large language model for integrating remote sensing and street-view imagery for understanding urban environments. To fuse satellite and street-view images, a cross-view perceiver is designed to enable the complementary fusion of cross-view knowledge and get semantically rich urban imagery feature. The strength of this paper includes rich datasets, impressive performance and diverse task coverage. The weakness is that the novelty of the proposed perceiver module is quite limited and the model comparison is insufficient. Therefore, I recommend another round of revision to incorporate the more comparison to show a comprehensive investigation on Urban Understanding.

**Additional Comments On Reviewer Discussion:**

The concerns raised by reviewers focus on details and quality of the collected dataset, the novelty of the proposed perceiver module, the fairness of the comparison experiment, and insufficient model comparison, inaccurate description of prior works and experimental analysis etc. The authors provided details of the proposed dataset, more experimental comparison and discussion about the technical contribution of the perceive in the rebuttal period. One reviewer ( iqRR)raised his rating from 3 to 6, two reviewers maintained their ratings, and one reviewer didn’t provide his final rating. In my opinion, there are some prior works (e.g.CrossFormer, DualFormer) focusing on fuse different views or modals, and the advantages of the proposed perceiver over these existing methods should be verified. Besides, it would be better if the proposed method is tested on prior benchmark Urbench .

---

### Decision · Program_Chairs · 2025-01-22

Reject